# TENSOR PRODUCT REPRESENTATION FOR MULTIVARIATE TIME SERIES FORECASTING

## ABSTRACT

Real-world multivariate time series exhibit nonstationary inter-variable dependencies, which evolve dynamically due to external environmental shifts. While capturing these intricate dynamics is crucial for accurate forecasting, many existing methods still struggle to explicitly model these complex relationships. This motivates the need for compositional learning, which explicitly separates relational and temporal components and flexibly recombines them. Such a design allows models to adapt to time-varying inter-variable relationships and generalize to unseen patterns. To address this, we introduce TS-TPR, a novel framework that employs tensor product representations for compositional learning. Specifically, context-aware role generation identifies the most salient relationships at each time, while hierarchical filler extraction summarizes the corresponding temporal patterns. By combining these dynamically generated roles and fillers via tensor products, TS-TPR creates an explicit, structured representation that naturally scales to many variables and adapts as dependencies shift. Through experiments on diverse real-world benchmarks, we show that TS-TPR not only outperforms state-of-the-art baselines but also provides interpretable, time-varying insights into inter-series interactions.

## 1 INTRODUCTION

Multivariate time series forecasting (MTSF) has gained increasing attention due to its broad applicability in domains such as finance (Huang et al., 2023a), climate (Wu et al., 2023), and transportation (Cheng et al., 2020). In practice, real-world time series often exhibit complex inter-variable relationships, including lead–lag effects, synchronous correlations, and indirect interactions. A key challenge in forecasting lies in the fact that these relationships are not fixed but evolve over time. Such dynamics are clearly illustrated in the Weather dataset (Wu et al., 2021), where carbon dioxide ($CO_2$) and water vapor concentration ($H_2OC$) exhibit an inverse relationship on cold winter mornings due to weak photosynthesis (Figure 1(a)), but synchronize on summer mornings when higher temperatures activate photosynthesis (Figure 1(b)). This example demonstrates that both the relevance of variables and the temporal patterns to be leveraged can vary depending on the underlying relationships.

However, most existing MTSF models implicitly encode inter-variable relationships while entangling them with temporal dynamics, which makes them vulnerable to distribution shifts when relational structures change between training and test phases. To improve robustness, it is essential to explicitly disentangle relational structures from temporal attributes, enabling the model to detect structural changes and adapt by extracting temporal patterns in novel relational contexts. This idea aligns with the notion of compositional generalization, where unseen cases can be understood by systematically recombining known components (Lake & Baroni, 2018; Webb et al., 2020). The tensor product representation (TPR) framework (Smolensky, 1990) provides a principled way to achieve such compositionality by binding a *role* vector, which encodes relational or structural patterns, with a *filler* vector, which captures content-specific attributes. This role–filler binding enables explicit structural representation and flexible recombination of learned components, and has shown promising generalization in tasks such as associative reasoning, mathematics, and natural language processing (Schlag & Schmidhuber, 2018; Li et al., 2019; Shi et al., 2022).

In the context of multivariate time-series forecasting, inter-variable relationships can naturally be represented as roles, while temporal characteristics serve as fillers. However, applying TPR to time

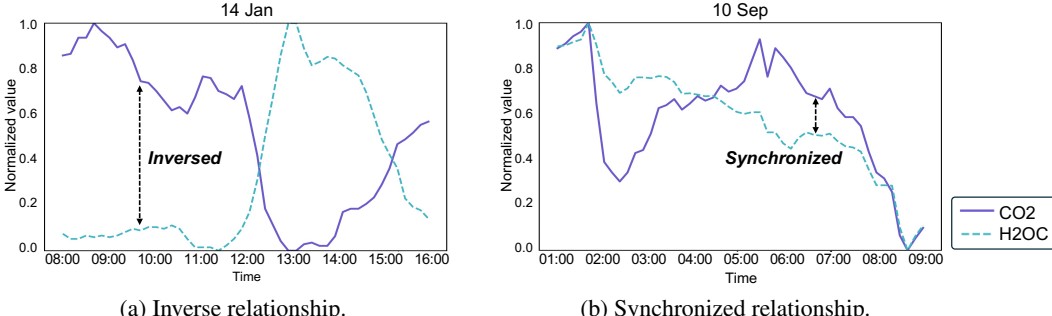

(a) Inverse relationship.         (b) Synchronized relationship.

Figure 1: Time-varying relationships between two variables in the Weather dataset: *Carbon dioxide* (CO2) and *Concentration of water vapor* (H2OC).

series introduces two key challenges. First, inter-variable relationships evolve dynamically over time, requiring roles to adapt to temporal dependencies and context-specific interactions, unlike the fixed roles seen in prior TPR applications (*e.g.,* syntactic roles in NLP). Second, as the number of variables increases, the space of possible relationships grows combinatorially, making it infeasible to explicitly define each as a separate role. These challenges necessitate learning generalized relational types that can flexibly capture variable interactions within the TPR framework, thereby extending its compositional principles to the dynamic and large-scale setting of multivariate time-series forecasting.

To address these challenges, we propose TS-TPR, a TPR-based framework tailored for MTSF. Instead of relying on fixed roles, TS-TPR introduces dynamically adaptable roles that capture context-dependent inter-variable relationships, while fillers encode the temporal attributes of each variable. To manage the combinatorial growth of dependencies, pairwise interactions are quantized into compact codebooks that serve as generalized relational types. This design enables each variable to ground its prediction on the most relevant relational role from the learned pool, allowing the model to flexibly reuse relational abstractions while adapting to structural changes over time. By explicitly disentangling relational structures from temporal attributes, TS-TPR leverages the strengths of structured representations while remaining responsive to the dynamics of time series data. As a result, the framework not only adapts to new relational contexts but also provides interpretability by revealing which relationships guide each prediction, and it further offers scalability by allowing the relational pool to expand naturally to broader forecasting scenarios. Here we summarize our key contributions as follows:

- We are the first to introduce TPR-based compositional learning for multivariate time series forecasting, enabling structured modeling of inter-variable relationships and temporal patterns.
- We propose TS-TPR, a TPR-based forecasting model that restructures roles and fillers to suit the characteristics of multivariate time series, enabling explicit and time-adaptive modeling of inter-variable dependencies.
- We demonstrate the performance and interpretability of TS-TPR through extensive experiments on various benchmark datasets, achieving state-of-the-art performance and analyzing relation type codes.

## 2 RELATED WORK

**Inter-Variable Modeling in Multivariate Time Series Forecasting**  Accurate multivariate time series forecasting (MTSF) requires effectively modeling inter-variable dependencies. Transformer-based methods such as TimeXer (Wang et al., 2024), iTransformer (Liu et al., 2023), and Crossformer (Zhang & Yan, 2022) capture these dependencies through attention mechanisms but often intertwine temporal patterns with relational information in a shared latent space, limiting interpretability and adaptability. Graph-based approaches like MTGNN (Wu et al., 2020) and Cross-GNN (Huang et al., 2023b) encode relations via edges, yet usually assume static or slowly varying structures, making it difficult to handle rapidly evolving dynamics. Recent advances, including At-

tnEmbed (Niu et al., 2024) and Abstractor (Altabaa et al., 2023), demonstrate that disentangling relational information from temporal attributes improves both generalization and interpretability. Motivated by these findings, we leverage attention maps to explicitly represent relational structures apart from temporal information, and employ TPR to structurally capture their interaction, enabling robust and interpretable forecasting under dynamic conditions.

**Tensor Product Representation (TPR)** TPR (Smolensky, 1990) is a neuro-symbolic framework that encodes structured information through the compositional binding of *role* and *filler* vectors. Each role $\mathbf{r}_i \in \mathbb{R}^d$ captures structural information (*e.g.,* position, grammatical role), while each filler $\mathbf{f}_i \in \mathbb{R}^d$ provides the content. The overall representation is constructed by summing tensor (outer) products of role–filler pairs,

$$\mathbf{M} = \sum_i \mathbf{r}_i \otimes \mathbf{f}_i, \tag{1}$$

where memory $\mathbf{M} \in \mathbb{R}^{d \times d}$ is the tensor–product representation matrix encoding all role–filler bindings. The content for a specific role can later be retrieved using an *unbinding vector* $\mathbf{u}_j$ via inner product: $\hat{\mathbf{f}}_j = \mathbf{M} \cdot \mathbf{u}_j$, assuming that $\mathbf{u}_j$ is approximately dual to $\mathbf{r}_j$.

While TPRs offer a principled way to represent compositional structure, their strict algebraic constraints are difficult to satisfy in MTSF settings, where dynamically evolving dependencies make roles and fillers hard to distinguish. Inspired by Soft TPR (Sun et al., 2024), which relaxes the canonical TPR specification by allowing representations to lie within an $\epsilon$-neighborhood of valid TPRs rather than strictly adhering to the exact role–filler form, we adopt a similar relaxation. This enables the model to capture quasi-compositional structures, such as approximate bindings or multiple fillers influencing a single role, while preserving the unbinding property. In TS-TPR, this relaxation is critical for handling dynamic and noisy inter-variable dependencies, as it permits flexible, distributed, and context-aware role–filler representations that approximate valid TPRs without being confined to rigid algebraic constraints.

**Problem Definition.** We consider a multivariate time series $\boldsymbol{\mathcal{X}} = \mathcal{X}^1, \dots, \mathcal{X}^N$ with $N$ variables. Given historical observations $\mathbf{X} = \boldsymbol{\mathcal{X}}_{t-T+1:t} \in \mathbb{R}^{N \times T}$ over a lookback window of length $T$, the goal is to predict future values $\mathbf{Y} = \boldsymbol{\mathcal{X}}_{t+1:t+F} \in \mathbb{R}^{N \times F}$ over a forecast horizon $F$.

## 3 PROPOSED METHOD: TS-TPR

We propose TS-TPR, a novel framework for multivariate time series forecasting (MTSF) inspired by TPR. Unlike simply adopting TPR, our approach redefines its core components to reflect the unique characteristics of multivariate time series. The key idea is to construct structured representations by disentangling relational structures from temporal attributes and expressing them as dynamically adaptable roles and fillers conditioned on temporal context. This enables TS-TPR to integrate structured relational information effectively, improving both generalization and interpretability.

### 3.1 OVERVIEW

As illustrated in Figure 2, our framework leverages two specialized embeddings to disentangle inter-variable dependencies from the temporal characteristics (§3.2). These embeddings are used to construct two core TPR components—*roles* and *fillers*—that are tailored for MTSF (§3.3). We then implement binding and unbinding operations using Linear Transformer-based attention mechanisms (§3.4). Finally, the retrieved fillers are used to generate the final predictions, with the training procedure described in (§3.5).

### 3.2 DISENTANGLED EMBEDDING

**Inter-variable Relation Embedding** We explicitly model inter-variable relations by quantifying the directed dependencies among different variables. Given a multivariate time series with $N$ variables, we embed time series $\mathbf{x}_i \in \mathbb{R}^T$ into a $d$-dimensional variate token $\mathbf{z}_i \in \mathbb{R}^d$. These $N$ variate tokens are fed into a shared multi-head self-attention module, which produces attention matrices $\mathbf{A}^{(l,h)} \in \mathbb{R}^{N \times N}$ for layer $l \in \{1, \dots, L\}$ and head $h \in \{1, \dots, H\}$. Since these matrices are computed from self-attention scores, they explicitly encode structural dependencies among variables

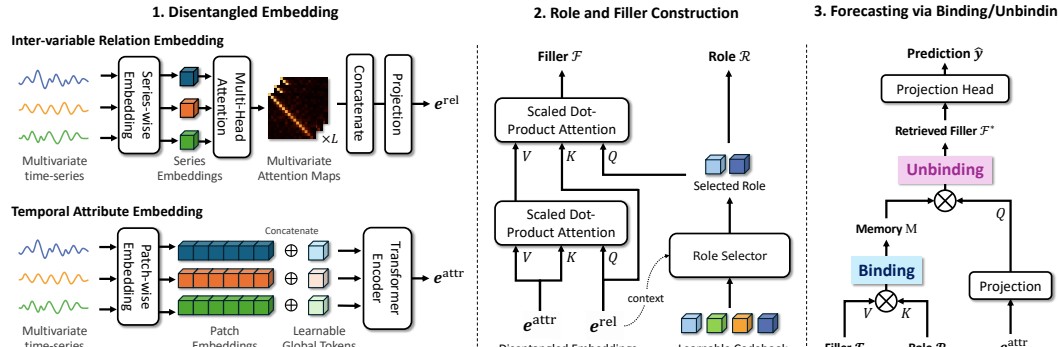

Figure 2: Overall framework of TS-TPR. The model disentangles inter-variable relation and temporal attribute embeddings (Step 1), constructs roles and fillers via context-aware role generation and hierarchical filler extraction (Step 2), and produces forecasts using retrieved fillers through TPR binding and unbinding (Step 3).

while focusing purely on relational interactions, without entangling the actual magnitudes of the variables. We stack all attention matrices and apply a linear projection to obtain the *inter-variable relation embedding*:

$$\mathbf{e}^{\text{rel}} = \text{Linear}([\mathbf{A}^{(1,1)}, \dots, \mathbf{A}^{(L,H)}]) \in \mathbb{R}^{N \times N \times d_{\text{rel}}}. \tag{2}$$

The relation embedding associated with target variable $i$ is denoted as $\mathbf{e}_i^{\text{rel}} \in \mathbb{R}^{N \times d_{\text{rel}}}$.

**Temporal Attribute Embedding**   Time series data often lack semantic structure at the level of individual time steps. To address this, each series $\mathbf{x}_i \in \mathbb{R}^T$ is divided into $P = \lfloor T/S \rfloor$ non-overlapping patches of length $S$. Each patch is then projected into a latent space via a linear layer, yielding patch embeddings $\mathbf{p}_i \in \mathbb{R}^{P \times d_{\text{attr}}}$, which are used to model temporal dependencies at the patch level, following Nie et al. (2022). We append a learnable global token $\mathbf{g}_i \in \mathbb{R}^{1 \times d_{\text{attr}}}$ to capture global contextual information, following the approach proposed in Wang et al. (2024). The resulting input sequence is then passed through a vanilla Transformer encoder. The final output, termed the *temporal attribute embedding*, is defined as:

$$\mathbf{e}^{\text{attr}} = \left\{ \text{Enc}([\mathbf{p}_i^1; \dots; \mathbf{p}_i^P; \mathbf{g}_i]) \right\}_{i=1}^N \in \mathbb{R}^{N \times (P+1) \times d_{\text{attr}}}. \tag{3}$$

Here, Enc denotes the Transformer encoder, and $\mathbf{e}_i^{\text{attr}} \in \mathbb{R}^{(P+1) \times d_{\text{attr}}}$ denotes the temporal attribute embedding for $i$-th target variable, consisting of both local-level and global-level token representations.

### 3.3   ROLE AND FILLER CONSTRUCTION FOR MTSF

**Context-aware Role Generation**   From the perspective of TPR, it is natural to represent roles as *inter-variable relationships* in MTSF. However, explicitly enumerating all possible pairwise relationships as role vectors is infeasible. To address this, we introduce a learnable codebook $\mathbf{C} = \{\mathbf{c}_k\}_{k=1}^K$, where each entry $\mathbf{c}_k \in \mathbb{R}^{d_{\text{rel}}}$ serves as a *role candidate*. For each relational embedding $\mathbf{e}_{i,j}^{\text{rel}}$, we map it to the closest codebook entry by solving and update the codebook by minimizing the vector quantization (VQ) loss:

$$k^* = \arg\min_k \left\| \mathbf{e}_{i,j}^{\text{rel}} - \mathbf{c}_k \right\|_2^2, \quad \mathcal{L}_{\text{VQ}} = \left\| \text{sg}[\mathbf{e}_{i,j}^{\text{rel}}] - \mathbf{c}_{k^*} \right\|_2^2 + \left\| \mathbf{e}_{i,j}^{\text{rel}} - \text{sg}[\mathbf{c}_{k^*}] \right\|_2^2, \tag{4}$$

where $\text{sg}[\cdot]$ denotes the stop-gradient operator. This objective ensures that the selected codebook entry $\mathbf{c}_{k^*}$ gradually converges to a representative prototype of recurrent relational patterns, while the embedding $\mathbf{e}_{i,j}^{\text{rel}}$ is pulled towards the closest code.

Once codebook mappings are obtained, the relational role set for target $i$ is defined by aggregating the $M$ closest codes with respect to the overall distance:

$$D_{i,k} = \sum_{j=1}^N \left\| \mathbf{e}_{i,j}^{\text{rel}} - \mathbf{c}_k \right\|_2^2, \quad \mathcal{R}_i = \{\mathbf{c}_{k_m^*}\}_{m=1}^M. \tag{5}$$

Here, $k_m^*$ denotes the index of the $m$-th selected code, which represents the relational context of $i$ by minimizing $D_{i,k}$. This formulation defines roles as generalized relational types while enabling their dynamic selection based on the underlying inter-variable dependencies.

**Hierarchical Filler Extraction** In order to effectively extract fillers corresponding to TPR roles, we propose a hierarchical filler extraction through a two-stage attention process. The *relation-aware fillers* locally select appropriate temporal attributes for each individual relation. In contrast, the *role-aware filler* aggregates these local selections from a global, role-specific perspective. This hierarchical extraction process ensures that the final filler representation captures both fine-grained, relation-specific temporal details and broader structural contexts defined by roles.

For the *relation-aware fillers*, we perform attention using relation embedding $\mathbf{e}_{(i,j)}^{\text{rel}}$ as a query, and temporal attribute embeddings of source variables $\mathbf{e}_j^{\text{attr}}$ as keys and values:

$$\mathcal{F}_{i,j}^{\text{rel}} = \text{Attention}(\mathbf{e}_{(i,j)}^{\text{rel}}, \mathbf{e}_j^{\text{attr}}, \mathbf{e}_j^{\text{attr}}) \in \mathbb{R}^{d_{\text{rel}}}, \tag{6}$$

where $\mathbf{e}_{(i,j)}^{\text{rel}} \in \mathbb{R}^{1 \times d_{rel}}$, $\mathbf{e}_i^{\text{attr}} \in \mathbb{R}^{(P+1) \times d_{attr}}$. the temporal attributes emphasized by specific inter-variable relations from source variables $x_j$ to a target variable $x_i$. However, each target variable typically interacts with multiple source variables, each emphasizing different temporal characteristics. As a result, relying solely on individual relationships makes it difficult to determine which temporal attributes should be prioritized within a given role.

To address this challenge, in the second stage of our hierarchical approach, we compute the final filler $\mathcal{F}_i$, referred to as *role-aware fillers*, by aggregating previously extracted fillers based on the target-specific roles. Specifically, we perform a second attention operation by using the role embedding $\mathcal{R}_i$ assigned to the target variable $x_i$ as a query, and relation embeddings corresponding to the target variable $\mathbf{e}_i^{\text{rel}}$ as keys. This attention identifies which relational instances are most relevant for each given role. The resulting attention weights are then applied to combine the relation-aware fillers associated with a target variable $\mathcal{F}_i^{\text{rel}} = [\mathcal{F}_{i,j}^{\text{rel}}]_{j=0}^N$, yielding a unified filler representation for each role:

$$\mathcal{F}_i = \text{Attention}(\mathcal{R}_i, \mathbf{e}_i^{rel}, \mathcal{F}_i^{rel}) \in \mathbb{R}^{N \times d_{\text{rel}}}. \tag{7}$$

## 3.4 FORECASTING VIA BINDING AND UNBINDING

Recently, Schlag et al. Schlag et al. demonstrated the equivalence between TPR and linear attention mechanisms, showing that keys, values, and queries correspond to roles, fillers, and unbinding operators, respectively. Building on this equivalence, the Linear Transformer integrates TPR's binding and unbinding operations to facilitate structural learning. urthermore, leveraging the Linear Transformer architecture ensures high computational efficiency, allowing scalability to long sequences and large multivariate settings. In this work, we introduce a novel *context-aware unbinding operator*, $\mathbf{u}^{\text{context}}$, Specifically, given temporal attribute embeddings corresponding to $i$th variable $\mathbf{e}_i^{\text{attr}} \in \mathbb{R}^{(P+1) \times d_{attr}}$, we first flatten these embeddings across the temporal dimension and subsequently apply a linear transformation to generate the context-aware unbinding operator for variable $i$:

$$\mathbf{u}_i^{\text{context}} = \text{Linear}\left(\text{Flatten}\left(\mathbf{e}_i^{\text{attr}}\right)\right). \tag{8}$$

Using the context-aware unbinding operator $\mathbf{u}_i \in \mathbb{R}^{M \times d_{attr}}$, role embeddings $\mathcal{R}_i \in \mathbb{R}^{M \times d_{attr}}$, and filler embeddings $\mathcal{F}_i \in \mathbb{R}^{M \times d_{attr}}$, we compute the retrieved filler $\mathcal{F}_i^*$ via linear attention:

$$\mathcal{F}_i^* = \text{LinearAttention}(\mathbf{u}_i, \mathcal{R}_i, \mathcal{F}_i). \tag{9}$$

The retrieved filler, unlike fixed operators that extract fillers tied to a specific role, adapts to the input by projecting into the role space, thereby exploring relational structures and dynamically extracting fillers aligned with them. The retrieved fillers $\mathcal{F}_i^*$, obtained by binding roles with fillers and applying the context-aware unbinding operators, are then passed into a learnable projection head $f_\theta$ to generate the final forecasts for each target variable $x_i$:

$$\hat{y}_i = f_\theta\left(\mathcal{F}_i^*\right). \tag{10}$$

### 3.5 JOINT OPTIMIZATION FOR TRAINING

The total loss combines the prediction loss $\mathcal{L}_{\text{pred}}$, defined as the mean squared error (MSE) between the ground-truth sequence $\mathbf{Y}$ and prediction $\hat{\mathbf{Y}}$, with a vector quantization loss $\mathcal{L}_{\text{VQ}}$ (see Eq. 4), that ensures the selected codes $\mathbf{c}^*$ remain meaningful and representative by jointly updating the codebook entries during training. The two components are weighted by $\alpha \in [0.2, 1]$ and $\beta \in [0, 0.5]$ to balance prediction accuracy and codebook regularization:

$$\mathcal{L}_{\text{total}} = \alpha \, \mathcal{L}_{\text{pred}} + \beta \, \mathcal{L}_{\text{VQ}}, \tag{11}$$

## 4 EXPERIMENTS

In this section, we evaluate the performance of the TS-TPR framework on multivariate time series forecasting. We begin by outlining the experimental setup (§4.1), followed by presenting the key results that demonstrate its effectiveness of TS-TPR across various forecasting tasks (§4.2). We then present ablation studies validating the framework's robustness and the contribution of its modules (§4.3). We conclude with an examination of its interpretability (§4.4) and anlayses, including scalability (§4.5) .

### 4.1 EXPERIMENTAL SETUP

**Datasets**    For the long-term multivariate time series forecasting (MTSF) task, we conduct experiments on seven widely used real-world benchmark datasets: Weather, Traffic, Electricity (ECL), and the ETT series (ETTh1, ETTh2, ETTm1, ETTm2). For the short-term multivariate forecasting task, we use the electricity price forecasting (EPF) dataset (Olivares et al., 2023). Further details on the data description are also provided in Appendix A.1.

**Implementation Details**    We adopt Adam  (Kingma & Ba, 2014) optimizer with an L2 loss for training TS-TPR. All experimental results are averaged over three runs with different random seeds, and we report the mean values in the result tables. All experiments are conducted on a single NVIDIA GeForce RTX 3090 GPU. For long-term forecasting, we follow prior studies (Liu et al., 2023) and set the lookback window to $T = 96$ with forecast horizons $F = 96, 192, 336, 720$. For short-term forecasting, we adopt the standard protocol of NBEATSx (Lago et al., 2021), using a lookback window of 168 and a prediction horizon of 24. In addition, we fix the patch length as 24 without overlap. For zero-shot forecasting, We adhere to the setups in (Jin et al., 2023) for fair comparisons, using a lookback window of $T = 512$ and forecast horizons of $F = \{96, 192, 336, 720\}$. Hyperparameter details in Appendix A.2.

**Baselines and Evaluation Metrics**    To valid the effectiveness of TS-TPR, we compare its forecasting performance with seven state-of-the-art baselines: TimeXer (Wang et al., 2024), iTransformer (Liu et al., 2023), PatchTST (Nie et al., 2022), Crossformer (Zhang & Yan, 2022), DLinear (Zeng et al., 2023), TimesNet Wu et al. (2022), SCINet (Liu et al., 2022), Autoformer (Wu et al., 2021), TSMixer(Chen et al., 2023), GPT4TS (Zhou et al., 2023), Time-LLM (Jin et al., 2023) and LLM-Time (Gruver et al., 2023). For evaluating forecasting performance, we use the mean square error (MSE) and mean absolute error (MAE) metrics.

### 4.2 MAIN RESULTS

The comprehensive MTSF results for long-term, short-term forecasting, and zero-shot forecasting are listed in Table 1, Table 2, and Table 3. All experiments are conducted under the multi-to-multi setting. Detailed results for each individual prediction length are provided in the Appendix B.3.

**Long-term Forecasting Performance**    Table 1 summarizes the average results across multiple forecasting horizons on standard multivariate benchmarks. In long-term forecasting, TS-TPR consistently outperforms competing methods on the ETT, ECL, and Weather datasets. On the Traffic dataset, the improvement is less pronounced, likely due to its relatively stable inter-variable dependencies, which limit the benefits of dynamic role modeling. By centering on relational structures, TS-TPR not only leverages cross-variable information effectively but also incorporates structured

Table 1: Long-term multivariate forecasting results. We compare various competitive models under different prediction lengths, following the setting of TimeXer (Wang et al., 2024). Results are averaged from all forecast horizons $F \in \{96, 192, 336, 720\}$, with a fixed lookback window $T = 96$ for all baselines. The best results are highlighted in **bold**, and the second-best in underline.

| Models | TS-TPR | | TimeXer | | iTransformer | | PatchTST | | DLinear | | Crossformer | | TimesNet | | Autoformer | |
|---|---|---|---|---|---|---|---|---|---|---|---|---|---|---|---|---|
| Metric | MSE | MAE | MSE | MAE | MSE | MAE | MSE | MAE | MSE | MAE | MSE | MAE | MSE | MAE | MSE | MAE |
| ETTh1 | **0.435** | **0.437** | 0.437 | **0.437** | 0.454 | 0.447 | 0.469 | 0.456 | 0.456 | 0.452 | 0.529 | 0.522 | 0.458 | 0.450 | 0.496 | 0.487 |
| ETTh2 | 0.371 | 0.398 | **0.367** | **0.396** | 0.383 | 0.407 | 0.387 | 0.407 | 0.559 | 0.515 | 0.942 | 0.684 | 0.414 | 0.427 | 0.450 | 0.459 |
| ETTm1 | **0.380** | **0.396** | 0.382 | 0.397 | 0.407 | 0.420 | 0.399 | 0.410 | 0.403 | 0.407 | 0.512 | 0.496 | 0.400 | 0.406 | 0.588 | 0.517 |
| ETTm2 | **0.274** | **0.322** | **0.274** | **0.322** | 0.288 | 0.332 | 0.281 | 0.326 | 0.350 | 0.410 | 0.757 | 0.610 | 0.291 | 0.333 | 0.327 | 0.371 |
| ECL | **0.166** | **0.269** | 0.171 | 0.270 | 0.178 | 0.270 | 0.216 | 0.304 | 0.212 | 0.300 | 0.244 | 0.334 | 0.192 | 0.295 | 0.227 | 0.338 |
| Weather | **0.239** | **0.270** | 0.241 | 0.271 | 0.258 | 0.278 | 0.259 | 0.280 | 0.265 | 0.317 | 0.259 | 0.315 | 0.259 | 0.287 | 0.338 | 0.382 |
| Traffic | 0.471 | 0.289 | 0.466 | 0.287 | **0.428** | **0.282** | 0.481 | 0.304 | 0.625 | 0.383 | 0.550 | 0.304 | 0.620 | 0.336 | 0.628 | 0.379 |
| 1$^{st}$ Count | **10** | | 4 | | 2 | | 0 | | 0 | | 0 | | 0 | | 0 | |

role–filler representations, thereby enhancing both robustness and interpretability in multivariate forecasting tasks. Full results are listed in Table 8 in Appendix B.3.

**Short-term Forecasting Performance**
To further examine the generalizability of TS-TPR in complex real-world scenarios, we extended our study to a multivariate short-term electricity price forecasting task, where strong and evolving dependencies naturally exist among multiple variables. This setting reflects a realistic scenario in which inter-variable relationships play a critical role in shaping temporal dynamics. For a more effective comparison, we adopted channel-dependency models as

Table 2: Short-term multivariate forecasting results on EPF dataset. We evaluate all baselines using an input length of 168 and a prediction length of 24.

| Model | TS-TPR | | TimeXer | | iTransformer | | Crossformer | | TSMixer | |
|---|---|---|---|---|---|---|---|---|---|---|
| Metric | MSE | MAE | MSE | MAE | MSE | MAE | MSE | MAE | MSE | MAE |
| NP | **0.292** | **0.316** | 0.305 | 0.336 | 0.331 | 0.345 | 0.297 | 0.326 | 0.320 | 0.355 |
| PJM | **0.079** | **0.180** | 0.085 | 0.191 | 0.087 | 0.195 | 0.092 | 0.181 | 0.087 | 0.197 |
| BE | **0.142** | **0.164** | 0.147 | 0.170 | 0.164 | 0.191 | 0.151 | 0.167 | 0.161 | 0.179 |
| FR | **0.149** | **0.152** | 0.157 | 0.163 | 0.168 | 0.174 | 0.164 | 0.153 | 0.158 | 0.170 |
| DE | **0.184** | **0.241** | 0.192 | 0.249 | 0.214 | 0.268 | 0.203 | 0.243 | 0.196 | 0.255 |
| *Avg* | **0.169** | **0.211** | 0.177 | 0.222 | 0.193 | 0.234 | 0.181 | 0.214 | 0.184 | 0.231 |

baselines across diverse architectures, including both Transformer- and MLP-based approaches, since they are specifically designed to capture cross-variable interactions. As shown in Table 2, TS-TPR consistently delivers superior performance across all five datasets, outperforming diverse competitive baselines. Importantly, when variable dependencies are strong, explicitly modeling their interactions while accounting for temporal dynamics yields clear performance advantages, highlighting the necessity of capturing evolving relationships in practical forecasting tasks. Overall, these results emphasize the distinctive strengths of TS-TPR in dependency-rich scenarios and demonstrate the scalability of our compositional learning framework to diverse multivariate forecasting settings.

**Zero-shot Forecasting Performance**  We further evaluate the zero-shot learning performance of TS-TPR under a cross-domain adaptation framework. Specifically, we investigate how well the model, trained on a source-domain dataset without accessing any samples from the target domain, generalizes to a new target dataset. To this end, we conduct experiments on the ETT benchmark across various cross-domain scenarios, with the results summarized in Table 3. The findings show that TS-TPR achieves comparable or lower MSE and MAE than the most competitive baselines. Our results demonstrate that, unlike LLM-based methods (Jin et al. (2023); Gruver et al. (2023)), TS-TPR achieves comparable performance using only time-series data and a lightweight architecture, without relying on an LLM backbone or additional textual supervision. This design is substantially more cost-effective while effectively leveraging specific relational information for knowledge transfer and reasoning. This is enabled by the codebook, which learns diverse types of relational structures and adaptively selects the most relevant ones as a role according to the target domain and temporal context. Full results are provided in Table 9 in Appendix B.4.

Table 3: Zero-shot multivariate forecasting result on ETT datasets. Following the setting Jin et al. (2023), We evaluate all baselines using an input length of 512 for all baselines. Results are averaged from all forecast horizons $F \in \{96, 192, 336, 720\}$.

| Models | **TS-TPR** | | TIME-LLM | | LLM-Time | | GPT4TS | | DLinear | | PatchTST | | TimesNet | | Autoformer | |
|---|---|---|---|---|---|---|---|---|---|---|---|---|---|---|---|---|
| Metric | MSE | MAE | MSE | MAE | MSE | MAE | MSE | MAE | MSE | MAE | MSE | MAE | MSE | MAE | MSE | MAE |
| ETTm1 → ETTm2 | **0.173** | **0.260** | 0.268 | 0.320 | 1.867 | 0.869 | 0.313 | 0.348 | 0.335 | 0.389 | 0.296 | 0.334 | 0.322 | 0.354 | 0.469 | 0.484 |
| ETTm2 → ETTh2 | 0.362 | 0.406 | **0.354** | **0.400** | 0.992 | 0.708 | 0.435 | 0.443 | 0.455 | 0.471 | 0.409 | 0.425 | 0.435 | 0.443 | 0.423 | 0.439 |
| ETTm1 → ETTh2 | **0.366** | **0.408** | 0.381 | 0.412 | 0.992 | 0.708 | 0.433 | 0.439 | 0.464 | 0.475 | 0.439 | 0.438 | 0.457 | 0.454 | 0.470 | 0.479 |
| *Average* | **0.300** | **0.358** | 0.334 | 0.377 | 1.284 | 0.762 | 0.394 | 0.410 | 0.418 | 0.445 | 0.381 | 0.399 | 0.405 | 0.417 | 0.454 | 0.467 |

## 4.3 ABLATION STUDY

In TS-TPR, we define roles and fillers tailored to MTSF. Roles are determined through a role selector (*RS*), while fillers are extracted via a two-stage hierarchical attention process, *H1* (Eq. 6) and *H2* (Eq. 7), aligned with the selected roles. To validate the design of these TPR components, we perform a detailed ablation study by removing individual submodules from the full model (TS-TPR). For the ablation study, we evaluate five model variants: (i) **Base variant**: Excludes all submodules. Roles are taken directly from the

Table 4: Ablation Results of modules in TS-TPR related to role and filler under the long-horizon setting ($F = 720$). *RS* indicates the Role selector and *H1* and *H2* denotes two stages of hierarchical filler extraction.

| | Role ($\mathcal{R}$) | Filler ($\mathcal{F}$) | | ETTh1 | | ETTh2 | | ECL | |
|---|---|---|---|---|---|---|---|---|---|
| | *RS* | *H1* | *H2* | MSE | MAE | MSE | MAE | MSE | MAE |
| Base | – | – | – | 0.516 | 0.498 | 0.450 | 0.453 | 0.242 | 0.332 |
| *RS*-only | ✓ | – | – | 0.506 | 0.494 | 0.447 | 0.451 | 0.232 | 0.324 |
| *w/o H2* | ✓ | ✓ | – | 0.500 | 0.485 | 0.446 | 0.451 | 0.205 | 0.307 |
| *w/o H1* | ✓ | – | ✓ | 0.539 | 0.520 | 0.425 | 0.440 | 0.202 | 0.305 |
| **TS-TPR** | ✓ | ✓ | ✓ | **0.468** | **0.468** | **0.418** | **0.436** | **0.198** | **0.300** |

entire learned codebook $\mathbf{C}$ without top-$m$ selection, and fillers are set as the temporal attribute $\mathbf{e}_i^{\text{attr}}$. (ii) **RS-only variant**: Includes only the *RS* module. The selected codes $\mathcal{R}_i$ serve as roles, while fillers remain $\mathbf{e}_i^{\text{attr}}$. This indicates that roles and fillers are not explicitly associated as pairs. (iii) **w/o H2 variant**: Removes the *H2* module. Fillers are defined as $\mathcal{F}_{i,j}^{\text{rel}}$ obtained from the first-stage extraction in Eq. 6. (iv) **w/o H1 variant**: Removes the *H1* module. A modified filler $\mathcal{F}_i'$ is defined by replacing $\mathbf{e}_i^{\text{rel}}$ and $\mathcal{F}_i^{\text{rel}}$ in Eq. 7 with $\mathbf{e}_i^{\text{attr}}$. (v) **Full model (TS-TPR)**: Incorporates all submodules (*RS, H1, H2*).

As shown in Table 4, adaptively selecting the most relevant roles through the *RS* module, rather than using all codebooks, consistently yields substantial improvements across datasets. This highlights the importance of context-aware role generation for capturing dynamic inter-variable dependencies. From the perspective of filler extraction, removing *H1* leads to a significant performance drop on the ETTh1 dataset, indicating that extracting temporal attributes aligned with fine-grained relational contexts via *H1* is particularly crucial in this setting. In contrast, on the ETTh2 and ECL datasets, removing *H2* causes a more pronounced degradation, suggesting that extracting temporal attributes conditioned on target-specific roles through *H2* plays a more critical role in these cases. Ultimately, the full TS-TPR model, which integrates RS and both stages of filler extraction, achieves the best performance across all datasets, including ETTh1, ETTh2, and ECL. This confirms the necessity of role-conditioned filler extraction within the TPR-based compositional learning framework.

## 4.4 INTERPRETABILITY OF TS-TPR

To evaluate whether the roles selected from the learned codebook effectively capture inter-variable relationship types and adapt to temporal changes, we visualize the code numbers assigned to roles alongside the corresponding time series. Figure 3 presents a case study on the ETTh1 dataset, illustrating the relationships among four input variables representing transformer loads (`HUFL`, `HULL`, `MUFL`, `MULL`) and the target variable, oil temperature (`OT`).

**Base regime** (Figure 3(a)) shows strong synchronization within the pairs (`HUFL`, `MUFL`) and (`HULL`, `MULL`), while a clear inverse relationship emerges between the two groups. These patterns align with their semantic meanings: `HUFL` and `MUFL` denote High/Middle UseFul Loads ("useful" loads),

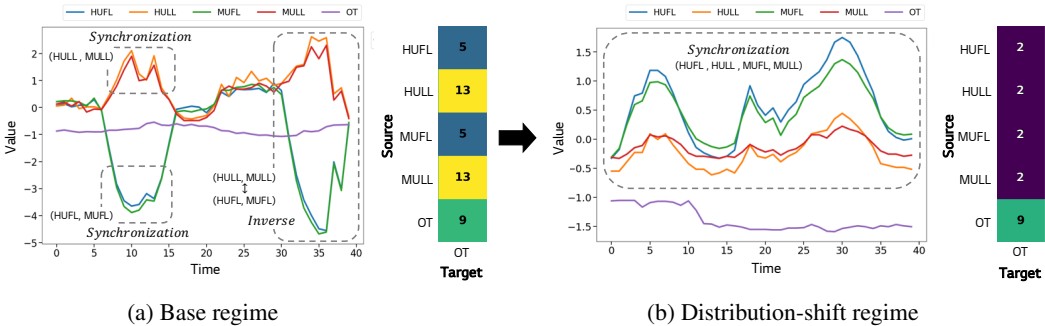

Figure 3: Visualization of the variates from ETTh1 dataset and the learned codebook under scenarios where inter-variable relationships dynamically change.

while HULL and MULL represent denote High/Middle UseLess Loads ("useless" loads), naturally leading to synchronized or inverse relational patterns. Correspondingly, when modeling the effect on OT, HUFL and MUFL are consistently mapped to code #5, while HULL and MULL are mapped to code #13. This demonstrates that the roles selected from the codebook successfully reflect the actual semantic relationships among variables.

**Distribution-shift regime** (Figure 3(b)) exhibits a transition in relational patterns: the previous inverse relationship between the two groups disappears, and all four variables become synchronized. Consequently, their influence on OT converges in the same direction, with all variables mapped to code #2. This demonstrates that the roles in the proposed framework capture stable dependencies under normal conditions while flexibly adapting to structural changes under distribution shifts, thereby enhancing interpretability and robustness in out-of-distribution scenarios.

### 4.5 FURTHER ANALYSIS OF TS-TPR

**Additional analyses (Appendix B.1)** First, we study scalability of TS-TPR under codebook size $K$ and selected roles $M$ (Figure 4), showing that compact configurations (e.g., $(K, M)=(16, 16)$ on ETTh1; $(64, 16)$ on Electricity) are sufficient despite combinatorially many potential relations. Second, we verify near-orthogonality of the learned role codebook via cosine-similarity heatmaps, supporting clean TPR encoding/decoding (Figure 5). Third, we examine the Role Selector and find an optimal $M$ that balances expressiveness and noise (e.g., $M=32$ on ETTh1, $M=8$ on Electricity); using the full codebook is not always beneficial (Figure 6).

**Computational complexity (Appendix B.2)** A concise complexity derivation for TS-TPR is provided in Table 6. In short, the overall cost is comparable to other SOTA models with an additional $\mathcal{O}(CM)$ from role selection ($C$ is number of variables, $M$ is role selection size). We also report empirical training efficiency on Weather (memory and ms/iter), showing that TS-TPR attains strong accuracy while remaining practically efficient relative to lightweight (PatchTST, iTransformer) and heavier (Crossformer) baselines (Table 7).

## 5 CONCLUSION

In this work, we introduced TS-TPR, extending Tensor Product Representation to multivariate time-series forecasting by explicitly disentangling relational roles from temporal fillers. Using codebook-based role generation, hierarchical filler extraction, and Linear-Transformer binding/unbinding, TS-TPR builds structured representations that adapt to changing inter-variable dependencies. Across diverse benchmarks, it outperforms strong baselines in both long- and short-term settings and proves effective in zero-shot cross-domain scenarios without any LLM backbone or textual inputs, making it cost-effective and practical. Beyond accuracy, TS-TPR offers interpretability, identifying which roles drive each prediction and how they shift under distribution change. This study establishes compositional TPR as a viable foundation for time-series forecasting and opens paths to scaling via a universal role codebook for broader, foundation-level models.

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

## A   Experimental Details

### A.1   Dataset Description

To evaluate the multivariate time series forecasting performance of the proposed TS-TPR, we conduct long-term forecasting experiments on seven real-world multivariate time-series datasets and short-term forecasting experiments on five real-world datasets. Table 5 provides a comprehensive overview of all benchmark datasets used in our experiments. For long-term forecasting, we use the following datasets: (1) ETT (Zhou et al., 2021): Consisting of four subsets — ETTh1 and ETTh2 (hourly records) and ETTm1 and ETTm2 (15-minute records). Each subset contains six electrical load variables and one oil temperature variable. (2) Weather (Zhou et al., 2021): Consisting of 21 meteorological variables recorded at 10-minute intervals by the Max Planck Institute for Biogeochemistry in 2020. (3) ECL (Li et al., 2019): Recording hourly electricity consumption for 321 clients. (4) Traffic (Wu et al., 2022): Logging hourly road occupancy rates from 862 highway sensors.

For short-term forecasting, we employ an electricity price forecasting dataset (Lago et al., 2021), covering six years and five distinct power markets: (1) NP: Nord Pool electricity market, with the hourly electricity price and corresponding grid load and wind power forecasts from Jan 1, 2013 to Dec 24, 2018. (2) PJM: Pennsylvania–New Jersey–Maryland market, recording the zonal electricity price in Commonwealth Edison (COMED), along with system load and COMED load forecasts from Jan 1, 2013 to Dec 24, 2018. (3) BE: Belgian electricity market, with the hourly electricity price and corresponding load forecasts in Belgium and generation forecasts from Jan 9, 2011 to Dec 31, 2016. (4) FR: French electricity market, including the hourly electricity price and corresponding load and generation forecasts from Jan 9, 2012 to Dec 31, 2017. (5) DE: German electricity market, containing the hourly electricity price, TSO Amprion load forecasts, and wind and solar generation forecasts from Jan 9, 2012 to Dec 31, 2017.

Table 5: Dataset descriptions. The Channel column specifies the number of variables, the Unit column denotes the time interval between consecutive observations, and the Dataset size is presented in the format (Train, Validation, Test).

| Dataset | Channel | Frequency | Dataset Size |
|---------|---------|-----------|--------------|
| ETTh1   | 7       | 1 Hour     | (8545, 2881, 2881)    |
| ETTh2   | 7       | 1 Hour     | (8545, 2881, 2881)    |
| ETTm1   | 7       | 15 Minutes | (34465, 11521, 11521) |
| ETTm2   | 7       | 15 Minutes | (34465, 11521, 11521) |
| Weather | 21      | 10 Minutes | (36792, 5271, 10540)  |
| ECL     | 321     | 1 Hour     | (18317, 2633, 5261)   |
| Traffic | 862     | 1 Hour     | (12185, 1757, 3509)   |
| NP      | 3       | 1 Hour     | (36500, 5219, 10460)  |
| PJM     | 3       | 1 Hour     | (36500, 5219, 10460)  |
| BE      | 3       | 1 Hour     | (36500, 5219, 10460)  |
| FR      | 3       | 1 Hour     | (36500, 5219, 10460)  |
| DE      | 3       | 1 Hour     | (36500, 5219, 10460)  |

### A.2   Hyperparameter Details

To support the experiments discussed in Section 4, we present the hyperparameter configuration used for training the proposed TS-TPR model. We consider the following key hyperparameters: latent dimension $d_{\text{latent}} \in \{128, 256, 512, 1024\}$, feature dimension $d_{\text{feat}} \in \{64, 128, 256, 512\}$, number of codebook entries $K \in \{4, 8, 16, 32, 64, 128\}$, number of selected roles $M \in \{4, 8, 16, 32, 64\}$, learning rate $lr \in \{0.001, 0.0001\}$, and dropout rate $dr \in \{0.1\}$. The batch size is varied over $\{2, 8, 16, 32\}$, while the number of encoder layers is fixed at 2, and the number of heads in multi-head attention is set to 8. For forecasting settings, we follow established protocols from prior work. In long-term forecasting, we adopt the TimeXer setting Wang et al. (2024) with input sequence

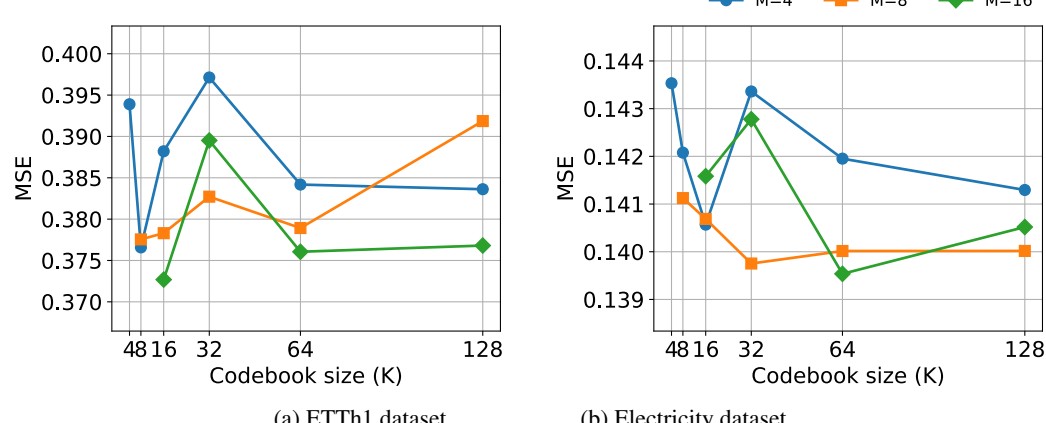

(a) ETTh1 dataset.       (b) Electricity dataset.

Figure 4: Forecasting results at horizon $H = 96$, varying role selection size $M$ and codebook size $K$ on the ETTh1 and Electricity datasets.

length $T \in \{96\}$ and forecast horizons $F \in \{96, 192, 336, 512\}$. For short-term forecasting, we use $T \in \{168\}$ with $F = 24$, following the same protocol. In the zero-shot forecasting setup of Time-LLM setting Jin et al. (2023), the input sequence length is set to $L = 512$ and the forecast horizons are $F \in \{96, 192, 336, 512\}$. This unified procedure ensures that our model is evaluated under diverse forecasting scenarios while maintaining comparability with strong baselines. Hyperparameter tuning is conducted using a grid search strategy that systematically explores the defined ranges. Model performance is evaluated using the Mean Squared Error (MSE), with the configuration yielding the lowest MSE selected for each experiment. This procedure ensures that the model is optimally configured for each dataset, thereby improving both forecasting accuracy and the robustness of the results.

# B EXPERIMENTAL RESULTS AND ANALYSIS

## B.1 ANALYSIS

**Efficient Scalability via Compact Role Codebooks** Figure 4 evaluates the scalability of TS-TPR by analyzing the effect of codebook size $K$ and the number of selected roles $M$ on forecasting accuracy. On the ETTh1 dataset (7 variables, 49 possible relations), the best performance is achieved with a compact configuration $(K, M) = (16, 16)$. In contrast, the Electricity dataset, with 321 variables and over 100k possible relations, attains peak accuracy with $(K, M) = (64, 16)$. These results indicate that even as the number of potential relations grows combinatorially, redundancy and similarity among relational patterns allow them to converge into a small, reusable set, confirming that our framework scales effectively to high-dimensional multivariate time series. Furthermore, selecting an appropriate $M$ reduces informational noise while maintaining efficiency and predictive accuracy. This ability to reuse a compact set of relation types ensures scalability and practical feasibility in high-dimensional settings. It also opens a pathway toward foundation models, where diverse relational types observed in multivariate time series can be pre-learned and adaptively reused for forecasting tasks.

**Orthogonality of Role Codebook** According to Smolensky Smolensky (1990), ensuring clear encoding and decoding within Tensor Product Representation (TPR) requires low correlation (orthogonality) among the roles. To verify this property, we visualize the cosine similarity heatmap of the learned role codebook embeddings in Figure 5. The results demonstrate that the proposed role codebook clearly distinguishes each role, similar to traditional TPR approaches, confirming that our proposed TS-TPR successfully achieves role orthogonality.

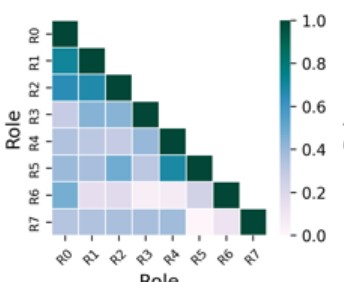

Figure 5: Visualization of the orthogonality of the role codebook in ETTh1 dataset.

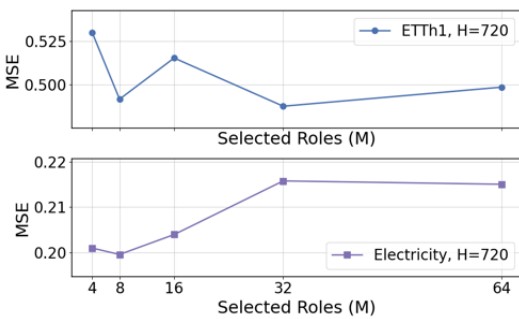

Figure 6: Forecasting MSE on ETTh1 (7 variables) and Electricity (321 variables) with a prediction horizon of $H = 720$, while varying the number of selected roles $M \in \{4, 8, 16, 32, 64\}$ under a fixed codebook size of $K = 64$.

**Effect of Role Selection**    We further analyze the impact of the number of selected roles $M$ while fixing the codebook size $K = 64$. Figure 6 shows the forecasting MSE on ETTh1 (7 variables) and Electricity (321 variables) datasets with a prediction horizon of $H = 720$. Interestingly, increasing $M$ does not always lead to better performance, even when the number of variables is large. For example, the best performance is achieved with $M = 32$ on ETTh1 and $M = 8$ on the Electricity dataset. This indicates that there exists an optimal value of $M$ that balances expressiveness and noise reduction. These results suggest that only a limited set of context-relevant roles is sufficient for accurate forecasting, validating the importance of the Role Selector in TS-TPR.

## B.2 Complexity Analysis

We provide a theoretical analysis of the computational complexity of TS-TPR, with look-back length $L$, prediction horizon $H$, number of variables $C$, model dimension $d$, patch length $P$, and role selection size $M$. The overall complexity can be expressed as $\mathcal{O}(C^2 + CM + CL + CH)$. Here, $\mathcal{O}(C^2)$ arises from constructing pairwise relation embeddings, $\mathcal{O}(C \cdot M)$ from selecting and processing the top-$M$ roles, $\mathcal{O}(C \cdot L)$ from temporal modeling, and $\mathcal{O}(C \cdot H)$ from projecting into the prediction horizon. As confirmed in Table 6, TS-TPR exhibits a similar complexity profile to other state-of-the-art (SOTA) models, with the main difference being the additional $\mathcal{O}(C \cdot M)$ term introduced by the role selection mechanism. Al-

Table 6: Complexity comparison of state-of-the-art time series forecasting models concerning window length $L$, number of channels $C$, forecasting horizon $H$, and role selection size $M$.

| Method | Complexity |
|---|---|
| TS-TPR (ours) | $\mathcal{O}(C^2 + CM + CL + CH)$ |
| iTransformer | $\mathcal{O}(C^2 + CL + CH)$ |
| TimeXer | $\mathcal{O}(C + (L/P)^2 + CH)$ |
| PatchTST | $\mathcal{O}(CL^2 + CH)$ |
| Transformer | $\mathcal{O}(CL + L^2 + HL + CH)$ |

though $\mathcal{O}(C^2)$ is theoretically required, its impact is mitigated in practice since only a relatively small number $M$ of roles are further processed, making the $\mathcal{O}(C \cdot M)$ overhead minor compared to the quadratic term. In addition, employing a Linear Transformer reduces the temporal cost from $\mathcal{O}((L/P)^2 \cdot d)$ in standard Transformers to $\mathcal{O}((L/P) \cdot d^2)$, ensuring efficiency for long look-back sequences.

We further compare the empirical training efficiency of TS-TPR with other SOTA models to validate our theoretical complexity analysis. As summarized in Table 7, TS-TPR (with $K = 64$ and $R = 8$) achieves the best forecasting performance (0.156) while keeping its memory footprint (1.82 GB) and training time (30.31 ms) at a moderate level. Although higher than lightweight baselines such as PatchTST or iTransformer, these costs remain substantially below heavier models like Crossformer. This confirms that the additional $\mathcal{O}(C \cdot M)$ term does not compromise efficiency, and TS-TPR provides a favorable balance between accuracy, relational modeling capacity, and computational cost.

Table 7: Comparison of training efficiency across multivariate time series forecasting models on the Weather dataset with an input length of 96 and a prediction length of 96. Reported metrics include training time (milliseconds per iteration, ms/iter) and memory footprint, measured as peak GPU memory usage (GB).

| Method | Memory Footprint (GB) | Time (ms) | MSE($\downarrow$) |
|---|---|---|---|
| PatchTST | 0.64 | 14.58 | 0.176 |
| Crossformer | 2.10 | 58.12 | 0.157 |
| iTransformer | 0.36 | 9.39 | 0.178 |
| TimeXer | 0.41 | 15.86 | 0.158 |
| TS-TPR | 1.82 | 30.31 | 0.154 |

## B.3 Full Results of Long-term Multivariate

Table 8 presents the full individual results of the long-term multivariate forecasting experiments on seven widely used benchmark datasets (Wu et al., 2022). Specifically, it expands Table 1 by reporting the separate results for all forecasting horizons $\{96, 192, 336, 720\}$ with the look-back window length fixed at 96. Each experiment is repeated three times with different random seeds, and the averaged results are presented. For comparison, we reference the results of prior methods, reported in (Wang et al., 2024; Han et al., 2024; Chen et al., 2025), which were conducted under the same experimental settings. Except for the Traffic dataset, our method consistently demonstrates superior performance across most benchmarks.

## B.4 Full Results of Zero-shot Performance

Table 9 presents the full results of zero-shot performance of TS-TPR under a cross-domain adaptation framework. Specifically, it expands Table 3 by reporting the separate results for all forecasting

horizon $\{96, 192, 336, 720\}$ with the look-back window length fixed at 96. Each experiment is repeated three times with different random seeds, and the averaged results are presented.

Table 8: Full Results of the long-term multivariate forecasting task. The best results are highlighted in **bold** and the second-best in underline.

| Models | | TS-TPR | | TimeXer | | iTransformer | | PatchTST | | Dlinear | | Crossformer | | TimesNet | | Autoformer | |
|---|---|---|---|---|---|---|---|---|---|---|---|---|---|---|---|---|---|
| Metric | | MSE | MAE | MSE | MAE | MSE | MAE | MSE | MAE | MSE | MAE | MSE | MAE | MSE | MAE | MSE | MAE |
| ETTh1 | 96 | **0.373** | **0.397** | 0.382 | 0.403 | 0.386 | 0.405 | 0.414 | 0.419 | 0.386 | 0.400 | 0.423 | 0.448 | 0.384 | 0.402 | 0.449 | 0.459 |
| | 192 | **0.428** | **0.431** | 0.429 | 0.435 | 0.441 | 0.436 | 0.460 | 0.445 | 0.437 | 0.432 | 0.471 | 0.474 | 0.436 | 0.429 | 0.500 | 0.482 |
| | 336 | 0.471 | 0.454 | **0.468** | **0.448** | 0.487 | 0.458 | 0.501 | 0.466 | 0.481 | 0.459 | 0.570 | 0.546 | 0.491 | 0.469 | 0.521 | 0.496 |
| | 720 | **0.468** | 0.468 | 0.469 | **0.461** | 0.503 | 0.491 | 0.500 | 0.488 | 0.519 | 0.516 | 0.653 | 0.621 | 0.521 | 0.500 | 0.514 | 0.512 |
| | Avg | **0.435** | **0.437** | 0.437 | **0.437** | 0.454 | 0.447 | 0.469 | 0.454 | 0.456 | 0.452 | 0.529 | 0.522 | 0.458 | 0.450 | 0.496 | 0.487 |
| ETTh2 | 96 | **0.286** | **0.338** | **0.286** | **0.338** | 0.297 | 0.349 | 0.302 | 0.348 | 0.333 | 0.387 | 0.745 | 0.584 | 0.340 | 0.374 | 0.346 | 0.388 |
| | 192 | 0.367 | 0.396 | **0.363** | **0.389** | 0.380 | 0.400 | 0.388 | 0.400 | 0.477 | 0.476 | 0.877 | 0.656 | 0.402 | 0.414 | 0.456 | 0.452 |
| | 336 | **0.412** | **0.423** | 0.414 | **0.423** | 0.428 | 0.432 | 0.426 | 0.433 | 0.594 | 0.541 | 1.043 | 0.731 | 0.452 | 0.452 | 0.482 | 0.486 |
| | 720 | 0.418 | 0.436 | **0.408** | **0.432** | 0.427 | 0.445 | 0.431 | 0.446 | 0.831 | 0.657 | 1.104 | 0.763 | 0.462 | 0.468 | 0.515 | 0.511 |
| | Avg | 0.371 | 0.398 | **0.367** | **0.396** | 0.383 | 0.407 | 0.387 | 0.407 | 0.559 | 0.515 | 0.942 | 0.684 | 0.414 | 0.427 | 0.450 | 0.459 |
| ETTm1 | 96 | **0.315** | **0.354** | 0.318 | 0.356 | 0.334 | 0.368 | 0.329 | 0.367 | 0.345 | 0.372 | 0.404 | 0.426 | 0.338 | 0.375 | 0.505 | 0.475 |
| | 192 | 0.364 | **0.383** | **0.362** | **0.383** | 0.387 | 0.391 | 0.367 | 0.385 | 0.380 | 0.389 | 0.450 | 0.451 | 0.374 | 0.387 | 0.553 | 0.496 |
| | 336 | **0.391** | **0.406** | 0.395 | 0.407 | 0.426 | 0.420 | 0.399 | 0.410 | 0.413 | 0.413 | 0.532 | 0.515 | 0.410 | 0.411 | 0.621 | 0.537 |
| | 720 | **0.449** | 0.442 | 0.452 | 0.441 | 0.491 | 0.459 | 0.454 | **0.439** | 0.474 | 0.453 | 0.666 | 0.589 | 0.478 | 0.450 | 0.671 | 0.561 |
| | Avg | **0.380** | **0.396** | 0.382 | 0.397 | 0.407 | 0.410 | 0.387 | 0.400 | 0.403 | 0.407 | 0.513 | 0.496 | 0.400 | 0.406 | 0.588 | 0.517 |
| ETTm2 | 96 | **0.169** | **0.254** | 0.171 | 0.256 | 0.180 | 0.264 | 0.175 | 0.259 | 0.193 | 0.292 | 0.287 | 0.366 | 0.187 | 0.267 | 0.255 | 0.339 |
| | 192 | **0.236** | **0.298** | 0.237 | 0.299 | 0.250 | 0.309 | 0.241 | 0.302 | 0.284 | 0.362 | 0.414 | 0.492 | 0.249 | 0.309 | 0.281 | 0.340 |
| | 336 | 0.297 | 0.339 | **0.296** | **0.338** | 0.311 | 0.348 | 0.305 | 0.343 | 0.369 | 0.427 | 0.597 | 0.542 | 0.321 | 0.351 | 0.339 | 0.372 |
| | 720 | 0.394 | 0.396 | **0.392** | **0.394** | 0.412 | 0.407 | 0.402 | 0.400 | 0.554 | 0.522 | 1.730 | 1.042 | 0.408 | 0.403 | 0.433 | 0.432 |
| | Avg | **0.274** | **0.322** | **0.274** | **0.322** | 0.288 | 0.332 | 0.281 | 0.326 | 0.350 | 0.401 | 0.757 | 0.610 | 0.291 | 0.333 | 0.327 | 0.371 |
| ECL | 96 | **0.139** | 0.243 | 0.140 | 0.242 | 0.148 | **0.240** | 0.195 | 0.285 | 0.197 | 0.282 | 0.219 | 0.314 | 0.168 | 0.272 | 0.201 | 0.317 |
| | 192 | 0.158 | 0.261 | **0.157** | 0.256 | 0.162 | **0.253** | 0.199 | 0.289 | 0.196 | 0.285 | 0.231 | 0.322 | 0.184 | 0.289 | 0.222 | 0.334 |
| | 336 | **0.171** | **0.274** | 0.176 | 0.275 | 0.178 | 0.269 | 0.215 | 0.305 | 0.209 | 0.301 | 0.246 | 0.337 | 0.198 | 0.300 | 0.231 | 0.338 |
| | 720 | **0.198** | **0.300** | 0.211 | 0.306 | 0.225 | 0.317 | 0.256 | 0.337 | 0.245 | 0.333 | 0.280 | 0.363 | 0.220 | 0.320 | 0.254 | 0.361 |
| | Avg | **0.166** | **0.269** | 0.171 | 0.270 | 0.178 | 0.270 | 0.216 | 0.304 | 0.212 | 0.300 | 0.244 | 0.334 | 0.192 | 0.295 | 0.227 | 0.338 |
| Weather | 96 | **0.154** | **0.201** | 0.157 | 0.205 | 0.174 | 0.214 | 0.177 | 0.218 | 0.196 | 0.255 | 0.158 | 0.230 | 0.172 | 0.220 | 0.266 | 0.336 |
| | 192 | **0.204** | 0.249 | **0.204** | **0.247** | 0.221 | 0.254 | 0.225 | 0.259 | 0.237 | 0.296 | 0.206 | 0.277 | 0.219 | 0.261 | 0.307 | 0.367 |
| | 336 | **0.261** | **0.290** | **0.261** | **0.290** | 0.278 | 0.296 | 0.278 | 0.297 | 0.283 | 0.335 | 0.272 | 0.335 | 0.280 | 0.306 | 0.359 | 0.395 |
| | 720 | **0.338** | **0.340** | 0.340 | 0.341 | 0.358 | 0.349 | 0.354 | 0.348 | 0.345 | 0.381 | 0.398 | 0.418 | 0.365 | 0.359 | 0.419 | 0.428 |
| | Avg | **0.239** | **0.270** | 0.241 | 0.271 | 0.258 | 0.279 | 0.259 | 0.281 | 0.265 | 0.317 | 0.259 | 0.315 | 0.259 | 0.287 | 0.338 | 0.382 |
| Traffic | 96 | 0.436 | 0.272 | 0.428 | 0.271 | **0.395** | **0.268** | 0.462 | 0.295 | 0.650 | 0.396 | 0.522 | 0.290 | 0.593 | 0.321 | 0.613 | 0.388 |
| | 192 | 0.459 | 0.283 | 0.448 | 0.282 | **0.417** | **0.276** | 0.466 | 0.296 | 0.598 | 0.370 | 0.530 | 0.293 | 0.617 | 0.336 | 0.616 | 0.382 |
| | 336 | 0.475 | 0.291 | 0.473 | 0.289 | **0.433** | **0.283** | 0.482 | 0.304 | 0.605 | 0.373 | 0.558 | 0.305 | 0.629 | 0.336 | 0.622 | 0.337 |
| | 720 | 0.515 | 0.311 | 0.516 | 0.307 | **0.467** | **0.302** | 0.514 | 0.322 | 0.645 | 0.394 | 0.589 | 0.328 | 0.640 | 0.350 | 0.660 | 0.408 |
| | Avg | 0.471 | 0.289 | 0.466 | 0.287 | **0.428** | **0.282** | 0.481 | 0.304 | 0.625 | 0.383 | 0.550 | 0.304 | 0.620 | 0.336 | 0.628 | 0.379 |
| 1st Count | | **17** | **14** | 10 | 11 | 4 | 6 | 0 | 1 | 0 | 0 | 0 | 0 | 0 | 0 | 0 | 0 |

Table 9: Full results of zero-shot multivariate forecasting experiment. Following the setting Jin et al. (2023), we adopt a fixed lookback window of $T = 512$ for all baselines and compare various competing models on the forecasting horizon of $F \in \{96, 192, 336, 720\}$.

| Models | | **TS-TPR** | | TIME-LLM | | LLMTime | | GPT4TS | | DLinear | | PatchTST | | TimesNet | | Autoformer | |
|---|---|---|---|---|---|---|---|---|---|---|---|---|---|---|---|---|---|
| Metric | | MSE | MAE | MSE | MAE | MSE | MAE | MSE | MAE | MSE | MAE | MSE | MAE | MSE | MAE | MSE | MAE |
| | 96 | 0.173 | 0.260 | **0.169** | **0.257** | 0.646 | 0.563 | 0.217 | 0.294 | 0.221 | 0.314 | 0.195 | 0.271 | 0.222 | 0.295 | 0.385 | 0.457 |
| | 192 | 0.228 | **0.300** | **0.227** | 0.318 | 0.934 | 0.654 | 0.277 | 0.327 | 0.286 | 0.359 | 0.258 | 0.311 | 0.288 | 0.337 | 0.433 | 0.469 |
| ETTm1 → ETTm2 | 336 | **0.282** | **0.335** | 0.290 | 0.338 | 1.157 | 0.728 | 0.331 | 0.360 | 0.357 | 0.406 | 0.317 | 0.348 | 0.341 | 0.367 | 0.476 | 0.477 |
| | 720 | **0.371** | 0.388 | 0.375 | **0.367** | 4.730 | 1.531 | 0.429 | 0.413 | 0.476 | 0.476 | 0.416 | 0.404 | 0.436 | 0.418 | 0.582 | 0.535 |
| | *Avg* | **0.264** | **0.321** | 0.268 | 0.320 | 1.867 | 0.869 | 0.313 | 0.348 | 0.335 | 0.389 | 0.296 | 0.334 | 0.322 | 0.354 | 0.469 | 0.484 |
| | 96 | 0.304 | 0.360 | **0.298** | **0.356** | 0.510 | 0.576 | 0.360 | 0.401 | 0.333 | 0.391 | 0.327 | 0.367 | 0.360 | 0.401 | 0.353 | 0.393 |
| | 192 | 0.369 | 0.403 | **0.359** | **0.397** | 0.523 | 0.586 | 0.434 | 0.437 | 0.441 | 0.456 | 0.411 | 0.418 | 0.434 | 0.437 | 0.432 | 0.437 |
| ETTm2 → ETTh2 | 336 | 0.383 | 0.425 | **0.367** | **0.412** | 0.640 | 0.637 | 0.460 | 0.459 | 0.505 | 0.503 | 0.439 | 0.447 | 0.460 | 0.459 | 0.452 | 0.459 |
| | 720 | **0.392** | 0.437 | 0.393 | **0.434** | 2.296 | 1.034 | 0.485 | 0.477 | 0.543 | 0.534 | 0.459 | 0.470 | 0.485 | 0.477 | 0.453 | 0.467 |
| | *Avg* | 0.362 | 0.406 | **0.354** | **0.400** | 0.992 | 0.708 | 0.435 | 0.443 | 0.455 | 0.471 | 0.409 | 0.425 | 0.435 | 0.443 | 0.423 | 0.439 |
| | 96 | **0.311** | 0.370 | 0.321 | **0.369** | 0.510 | 0.576 | 0.353 | 0.392 | 0.365 | 0.415 | 0.354 | 0.385 | 0.377 | 0.407 | 0.435 | 0.470 |
| | 192 | **0.359** | **0.397** | 0.389 | 0.410 | 0.523 | 0.586 | 0.443 | 0.437 | 0.454 | 0.462 | 0.447 | 0.434 | 0.471 | 0.453 | 0.495 | 0.489 |
| ETTm1 → ETTh2 | 336 | **0.383** | **0.418** | 0.408 | 0.433 | 0.640 | 0.637 | 0.469 | 0.461 | 0.496 | 0.494 | 0.481 | 0.463 | 0.472 | 0.484 | 0.470 | 0.472 |
| | 720 | 0.410 | 0.447 | **0.406** | **0.436** | 2.296 | 1.034 | 0.466 | 0.468 | 0.541 | 0.529 | 0.474 | 0.471 | 0.495 | 0.482 | 0.480 | 0.485 |
| | *Avg* | **0.366** | **0.408** | 0.381 | 0.412 | 0.992 | 0.708 | 0.433 | 0.439 | 0.464 | 0.475 | 0.439 | 0.438 | 0.457 | 0.454 | 0.470 | 0.479 |