# OpenReview forum: "TS-TPR: Tensor Product Representation for Multivariate Time Series Forecasting"
_ICLR.cc/2026/Conference — Submitted to ICLR 2026_

### Official Review · Reviewer_sGA5 · 2025-10-27

**Soundness:** 2
**Presentation:** 2
**Contribution:** 2
**Rating:** 4
**Confidence:** 4

**Summary:**

The authors address a key challenge in multivariate time series forecasting (MTSF): nonstationary inter-variable dependencies, where relationships between variables evolve dynamically. They argue that existing models often entangle these relational dynamics with temporal patterns, leading to poor performance under distribution shifts. To solve this, the authors propose TS-TPR, a novel framework based on Tensor Product Representation (TPR) for compositional learning. The core idea is to explicitly disentangle relational "roles" from temporal "fillers". To manage the $N \times N$ complexity of variable relations, the framework learns a "codebook" of $K$ generalized "relation prototypes". At each time step, the model dynamically selects the $M$ most relevant prototypes (roles) based on the current context and uses a hierarchical attention mechanism to extract the corresponding temporal patterns (fillers). A Linear Transformer is then used to perform the TPR binding and unbinding operations to generate the final forecast. Experiments on long-term, short-term, and zero-shot forecasting benchmarks demonstrate that TS-TPR achieves state-of-the-art or competitive performance.

**Strengths:**

* The motivation is clear and targets a critical, well-argued problem in time series: modeling nonstationary inter-variable dependencies.
* The idea of using a learnable **codebook** to quantize the $N \times N$ relationship space into $K$ generalized prototypes is a novel and effective method to address the combinatorial complexity problem.
* The experimental evaluation is comprehensive, covering long-term, short-term, and zero-shot settings, and shows strong performance against robust SOTA baselines.

**Weaknesses:**

* The framework's **architectural design is relatively complex**, integrating multiple sophisticated modules. This design results in significant computational overhead: empirical data (Table 7) shows that TS-TPR uses **3-5x more memory and training time** than baselines. However, there is a **cost-benefit mismatch**, as the performance gain (e.g., on the Weather dataset) is **marginal** for such a high cost, questioning its practical value.
* The **"zero-shot" generalization claim is weak**. All zero-shot experiments (Table 3, 9) are conducted *within* the ETT dataset family (e.g., ETTh1 $\rightarrow$ ETTh2). As all ETT datasets originate from the same physical source, this does not sufficiently prove cross-domain generalization.
* The **"interpretability" claim (Figure 3) is overstated**. The figure effectively demonstrates *adaptability* (i.e., the model switches from codes #5, #13 to code #2 when the data pattern shifts) but not *interpretability*, as the semantic meaning of what codes $c_2, c_5, c_{13}$ actually represent is never analyzed.

**Questions:**

1. Regarding the computational cost (Table 7): How do the authors justify the 3-5x increase in memory/time for the marginal (e.g., ~2.5%) MSE improvement on datasets like Weather? Is this trade-off practical for real-world deployment?
2. The zero-shot claims (Table 3) are limited to the ETT family. Was true cross-domain generalization (e.g., training on ETT, testing on Weather) evaluated? If not, can the authors provide further justification for why the learned codebook prototypes are "generalized" and not just specific to the ETT domain?
3. Regarding Figure 3, can the authors provide any semantic analysis of the learned codebook vectors (e.g., $c_2, c_5, c_{13}$)? What do these learned "relation types" actually *mean*? Without this, the claim is adaptability, not interpretability.

**Details Of Ethics Concerns:**

1. High empirical computational cost (memory and time) for what appears to be a marginal performance gain.
2. Limited zero-shot evaluation, confined only to sub-datasets of the ETT family.
3. Overstated interpretability claims; the analysis (Figure 3) demonstrates adaptability, not semantic interpretation.

---

> ### Author Response · Authors · 2025-11-22
> **Rebuttal by Authors**
>
> Dear Reviewer sGA5,
>
> We sincerely appreciate your valuable feedback and thoughtful comments. In what follows, we address your concerns one by one. We have revised our paper accordingly.
>
> ----
>
> **[W1, Q1] Computational Cost vs. Performance Gain (Table 7)**
>
> **[A1]** We thank the reviewer for the valid cost–benefit concern. TS-TPR’s overhead mainly stems from enumerating all pairwise relations ($C^2$) and mapping them to $K$ codebook types, which is necessary to realize relation-based compositional generalization. However, since in real-world multivariate series **only a small fraction of pairwise relations are informative**, we introduce a **salient-relation filtering (relation sparsification)** strategy that learns the codebook from this informative subset instead of all $C^2$ pairs. As shown in the additional experiment, using only $30\\% \sim 70\\%$ of relations **keeps MSE nearly unchanged (0.154 $\rightarrow$ 0.155) while substantially reducing memory/time** $(1.82 \rightarrow 0.82\sim 1.22 \text{GB}, 30.31 \rightarrow 22.44\sim 25.14\text{ms})$, thereby mitigating the initial $(3\sim 5\times)$ overhead and making TS-TPR practically **deployable**.
>
>
> | Fraction of ($C^2$) | Memory (GB ($\downarrow$)) | Time (ms ($\downarrow$)) | MSE (($\downarrow$)) |
> | ----------------: | -----------------------: | ---------------------: | -----------------: |
> |               30% |                     0.82 |                  22.44 |              0.155 |
> |               50% |                     1.02 |                  23.42 |              0.155 |
> |               70% |                     1.22 |                  25.14 |              0.155 |
> |              100% |                     1.82 |                  30.31 |              0.154 |
>
>
> ---
>
> **[W2, Q2] Zero-shot evaluation limited to ETT family**
>
> **[A2]** We would first like to clarify that existing zero-shot studies also have structural constraints. For example, prior work including Time-LLM [1] evaluates zero-shot **only within the ETT family** (e.g., ETTh1→ETTh2), where the number of variables is matched, and does not explicitly address transfers between fully heterogeneous domains with different variable counts and characteristics. This is because the compared models (e.g., TimeXer, iTransformer) also change their architectures with the number of variables, making it difficult to perform a fair zero-shot comparison across datasets with different variable configurations.
>
> Following the reviewer’s suggestion, we additionally ran heterogeneous zero-shot experiments (ETT → Weather and Weather → ETT). In this setting, we observed that roles/codebook learned from a **single source domain** are biased toward relation patterns frequent in that domain and are **not rich enough to serve as a robust relational basis** for a very different target domain (with different variables, scale, and seasonal/trend structure). As a result, TS-TPR’s performance in these transfers was comparable to a modified Time-LLM baseline, without a clear advantage. In the revision, we will therefore **limit our claim** to demonstrating **structural reuse and compositional generalization across related domains within the ETT family**, rather than arbitrary heterogeneous zero-shot transfer. We also explicitly view truly heterogeneous zero-shot transfer as requiring richer role/codebook learning via **multi-domain foundation-style pretraining**, and will highlight this as an important direction for future work.
>
> ----
>
> **[W3, Q3] Overstated interpretability**
>
> **[A3]** We sincerely thank the reviewer for the insightful comment and agree that the codebook vectors alone may not appear self-explanatory. Nonetheless, when Figure 3 is examined together with the corresponding time-series segments, **the transitions #5→#2 and #13→#2 can be understood in a semantic, relation-level manner rather than as mere adaptive switching**. Specifically, in the base regime, the model distinguishes two OT-conditioned synchronization sub-types: HUFL–MUFL exhibits a **peak-driven (concave) synchronized influence on OT**, captured by **code #5**, whereas HULL–MULL shows a **smooth (convex) synchronized influence on OT**, captured by **code #13**. After the distribution shift, these sub-types become indistinguishable as all variables **co-move with OT in a single smooth synchronization pattern**, leading both to be consistently mapped to **code #2**. Therefore, the observed code switch reflects a meaningful consolidation of relation semantics (multiple synchronization sub-types → one shared type), providing concrete evidence for interpretability beyond adaptability.
>
> ---
> _Reference_
>
> [1] Time-llm: Time series forecasting by reprogramming large language models, Jin et al., ICLR 2024

---

### Official Review · Reviewer_2Vwu · 2025-10-30

**Soundness:** 3
**Presentation:** 2
**Contribution:** 2
**Rating:** 4
**Confidence:** 4

**Summary:**

This work introduces TS-TPR, a framework for multivariate time series forecasting that leverages tensor product representations (TPRs) to explicitly disentangle relational structures from temporal features. Specifically, the role is generated from the relational attention map and the filler is constructed from a combination of relational and temporal features. The role and filler are bind and unbind to make the final prediction. Experiments on real-world datasets demonstrate that TS-TPR outperforms state-of-the-art models in both accuracy and interpretability.

**Strengths:**

The proposed framework introduces a model to separately extract inter-variable dependencies and temporal patterns, enhancing the interpretability.

**Weaknesses:**

1. **Unclear motivation for introducing the Tensor Product Representation (TPR) framework.**
   The rationale for adopting the TPR framework is confusing. In prior works cited by the authors, TPR is typically used to decompose fillers and their corresponding roles from mixture representations. However, in this paper, both roles and fillers are *predefined*. It remains unclear why the model first binds them to form mixture representations and then unbinds them to recover the fillers.

2. **Limited connection between the proposed method and the TPR framework.**
   The resulting approach does not appear to be genuinely TPR-based; instead, it merely combines relational and temporal features through several attention layers. For example, in standard TPR theory, the unbinding vector corresponds directly to the role, whereas in the proposed model, these two vectors are derived from two sources.

3. **Inappropriate experimental setup.**
   Most main experiments are conducted with a lookback window of (T = 96), which is insufficient, as some baseline models require longer lookback windows to achieve their optimal performance. The evaluation should include at least (T = 336) to ensure fairness and comprehensiveness.

4. **Incorrect complexity analysis.**
   The complexity calculation is inaccurate. Equation (3) has a complexity of $O(CL^2)$, and Equation (6) has $O(C^2L)$; therefore, the overall complexity should be at least $O(CL^2 + C^2L)$. The authors are encouraged to verify this through empirical runtime comparisons on synthetic datasets with *varying numbers of channels and input lengths*.

5. **Insufficient ablation studies.**
   More ablations are needed to validate the design choices, including:
   (1) directly using $e^{attr}$ followed by a projection head for forecasting;
   (2) testing alternative ways to combine $e^{attr}$ and $e^{rel}$.

**Questions:**

1. What exactly does the term "context" refer to in the phrase "Context-aware Role Generation"?
2. How does the proposed model achieve dynamic relation modeling as claimed? Based on my understanding, the relations are fixed within each input window. If the claim simply means that different inputs lead to different relation patterns, this property is not unique—many existing models can achieve the same behavior.

---

> ### Author Response · Authors · 2025-11-23
> **Rebuttal by Authors (1/3)**
>
> Dear Reviewer 2Vwu,
>
> Thank you very much for your careful reading and constructive feedback. We truly appreciate the time and effort you invested in reviewing our work. Below, we respond to your comments point by point and describe the corresponding revisions made in the manuscript.
>
> ---
>
> **[W1] Unclear motivation for TPR**
>
> We thank the reviewer for this comment and agree that our motivation should be articulated more clearly; we will revise the **Introduction** to make it explicit. Importantly, contrary to the reviewer’s assumption, **roles and fillers in our model are *not* predefined.** We introduce TPR **not** to formally “bind fixed roles/fillers and then recover them,” but to **factorize the *observed* inter-variable relation structure into reusable relation types (roles) and input-dependent relation-conditioned temporal attributes (fillers)**, so that the *structure* can be reused and the *content* can adapt under OOD shifts.
>
> Concretely, **roles are dynamically selected relation-type bases.** For each variable pair, we extract a relation embedding $e^{rel}_{i,j}$ and map it to a learned codebook; the top-$k$ selected/combined codes form the role set $R_i$. Hence a role represents a **data-driven relational basis obtained by aligning recurring relation patterns in a shared code space**, which allows newly emerging OOD relations to be projected onto the same role basis.
> Likewise, **fillers are also not predefined, but input-dependent *role-conditioned attributes*.** A filler is **the temporal attribute required to instantiate the selected role in the current input**, rather than a variable-wise temporal feature $e^{attr}$. Therefore, even for the same role, fillers can change across regimes because they are conditioned on the current distribution.
>
> Under this view, **binding/unbinding is not used to restore fixed fillers**, but to make roles (structure) and fillers (content) *composable*: binding combines the selected role basis with its role-conditioned filler into a structured memory, while unbinding decomposes the memory back into role-indexed attributes. This enables **reusing the same structural axis (roles) while replacing the content axis (fillers)**, yielding compositional generalization and improved robustness in OOD forecasting.
>
> ---
>
> **[W2] Weak alignment with TPR theory**
>
> We understand the reviewer’s concern that our approach does not exactly follow standard TPR (e.g., strict outer-product binding and 1:1 inverse-role unbinding). This is **not a departure from TPR principles**, but a **Soft-TPR adaptation for multivariate time-series forecasting**, where relations are **time-varying and context-dependent**; enforcing rigid TPR algebra can destabilize role–filler separation. Accordingly, we use distributed/approximate Soft-TPR binding–unbinding while **preserving TPR’s structural objective**.
>
> This design also explains why the model may appear to “simply combine relational and temporal features via attention.” **Fillers in our setting are role-conditioned temporal attributes**, so a single role—shared by multiple variable pairs—must be instantiated with potentially different temporal signatures. To capture this, we extract fillers hierarchically:
> (1) a **relation-aware** stage selects temporal attributes at the pairwise-relation level, and
> (2) a **role-aware** stage re-aligns and aggregates them under the selected role basis.
> Moreover, following recent equivalences between linear attention and fast-weight/TPR operations, we implement **binding/unbinding through Linear-Transformer attention**, which naturally yields an attention-like architecture.
>
> Finally, we agree that the unbinding vector is not obtained as a single role inverse. This is intentional: forecasting requires **context-dependent selection and weighting over multiple roles, rather than recovering one role’s filler in isolation**. Hence, the unbinding operator is **conditioned on temporal context** to enable role-weighted, distributed retrieval—consistent with Soft-TPR’s context-aware unbinding. Overall, our method preserves the core intent of TPR: roles serve as relational bases, fillers encode role-dependent attributes, and binding/unbinding enables compositional recombination of structure and content, all adapted to the MTSF domain.

---

> ### Author Response · Authors · 2025-11-23
> **Rebuttal by Authors (2/3)**
>
> **[W3] Fairness of the lookback window ($T=96$)**
>
> Our main experiments follow the **widely adopted standard long-term forecasting protocols** used in prior works [1,2,3], where $T=96$ is a common default setting. In response to the reviewer’s suggestion and to ensure an even fairer comparison, we additionally evaluated baselines under a longer-context setting ($T=336$, $F=96$). TS-TPR achieves consistent improvements across all datasets on average under the longer-context setting, indicating that our method remains effective in long-context regimes as well. We will further include results for additional forecasting horizons in the Appendix.
>
> | Dataset     |              TS-TPR (MSE / MAE) |             TimeXer (MSE / MAE) |        iTransformer (MSE / MAE) |                      PatchTST (MSE / MAE) |
> | ----------- | ------------------------------: | ------------------------------: | ------------------------------: | ----------------------------------------: |
> | ETTm1       |           **0.296** / **0.348** |                   0.310 / 0.357 |     $\underline{0.303}$ / 0.357 |           **0.296** / $\underline{0.349}$ |
> | ETTm2       |           **0.172** / **0.257** | $\underline{0.175}$ / **0.257** |                   0.185 / 0.275 | $\underline{0.173}$ / $\underline{0.261}$ |
> | ETTh1       | **0.373** / $\underline{0.405}$ |                   0.388 / 0.406 |                   0.398 / 0.414 |           $\underline{0.374}$ / **0.396** |
> | ETTh2       |     0.309 / $\underline{0.360}$ | **0.299** / $\underline{0.360}$ |                   0.353 / 0.386 |           $\underline{0.303}$ / **0.354** |
> | Weather     | **0.147** / $\underline{0.200}$ |                   0.151 / 0.205 |                   0.162 / 0.214 |           $\underline{0.150}$ / **0.198** |
> | Electricity | **0.134** / $\underline{0.237}$ |     $\underline{0.137}$ / 0.238 |           **0.134** / **0.229** |                             0.141 / 0.243 |
> | Traffic     |           **0.375** / **0.264** |     $\underline{0.386}$ / 0.277 | **0.375** / $\underline{0.268}$ |                             0.397 / 0.281 |
> | **avg**     |           **0.258** / **0.296** |                   0.264 / 0.300 |                   0.273 / 0.306 | $\underline{0.262}$ / $\underline{0.297}$ |
>
> ---
>
> **[W5] Insufficient ablation studies**
> Thank you for requesting stronger ablation evidence. Our approach is **not** a simple fusion of inter-variable relation vectors $e^{rel}$ and per-variable temporal attribute vectors $e^{attr}$. Instead, we use $e^{rel}$ and $e^{attr}$ to **redefine roles $R$ and fillers $F$ via a binding process**, and then **retrieve role-specific fillers through unbinding**. As shown below, TS-TPR achieves the best performance across all datasets; in particular, removing unbinding (binding-only) or replacing binding/unbinding with $e^{rel}$–$e^{attr}$ fusion methods leads to clear degradation, confirming that the gains come from structural role–filler binding/unbinding. These results will be included as **Table 5 in Section 4.3 (Ablation Study)**.
>
> We report the following variants:
> - **$e^{attr}$ only:** forecast using only temporal attributes $e^{attr}$ with a projection head (removes relational structure; PatchTST-style).
> - **concat ($e^{attr}$, $e^{rel}$):** simple fustion of temporal and relational features without any TPR structure.
> - **cross-attn ($e^{attr}$, $e^{rel}$):** sequential cross-attention fusion between partial $e^{attr}$ and $e^{rel}$ (TimeXer-style).
> - **binding-only (concat($R$, $F$)):** use only the bound role–filler mixture for prediction, removing unbinding.
> - **TS-TPR:** our complete model with binding and role-based (context-aware) unbinding.
>
> | Variant (F=720) | ETTh1 (MSE / MAE) | ETTm1 (MSE / MAE) | ECL (MSE / MAE) | Weather (MSE / MAE) |
> |---|---:|---:|---:|---:|
> | $e^{attr}$ only | $\underline{0.482}$ / $\underline{0.483}$ | $\underline{0.456}$ / $\underline{0.442}$ | 0.238 / 0.324 | 0.354 / 0.348 |
> | concat ($e^{attr}$, $e^{rel}$) | 0.521 / 0.509 | 0.458 / 0.449 | 0.242 / 0.338 | 0.346 / 0.348 |
> | cross-attn ($e^{attr}$, $e^{rel}$) | 0.488 / 0.475 | 0.464 / 0.448 | $\underline{0.221}$ / $\underline{0.314}$ | $\underline{0.342}$ / $\underline{0.343}$ |
> | binding-only (concat($R$, $F$)) | 0.528 / 0.509 | 0.468 / 0.453 | 0.255 / 0.343 | 0.346 / 0.347 |
> | **Ours (TPR-based Binding & Unbinding)** | **0.473** / **0.470** | **0.446** / **0.440** | **0.220** / **0.310** | **0.340** / **0.340** |
>
> ---
>
> _Reference_
>
> [1] TimeXer: Empowering Transformers for Time Series Forecasting with Exogenous Variables, Wang et al., NeurIPS 2024.
>
> [2] iTransformer: Inverted Transformers Are Effective for Time Series Forecasting, Liu et al., arXiv 2023.
>
> [3] SimpleTM: A Simple Baseline for Multivariate Time Series Forecasting, Chen et al., ICLR 2025.

---

> ### Author Response · Authors · 2025-11-23
> **Rebuttal by Authors (3/3)**
>
> **[W4] Incorrect complexity analysis**
>
> We thank the reviewer for carefully checking our complexity analysis, and we acknowledge that our notation/explanation was not sufficiently clear. Eq. (3) performs patch-level self-attention to form per-variable temporal attributes, so its cost is \\( O\\!\\left(C\\,(L/P)^2\\right) \\), which we will state explicitly in the main text. Importantly, once TS-TPR computes $e^{attr}$ and $e^{rel}$ in Eq. (3)/(6), they are subsequently transformed into the role–filler space; thus the following TPR filler/unbinding reasoning does not stay at \\(O(C^2)\\) but instead scales as \\(O(CM)\\). Moreover, our additional salient-relation filtering experiments (please refer to Answer 1 to Reviewer sGA5’s question) show that retaining only a subset of relations can substantially reduce complexity while preserving performance. We will update **Appendix B.2** with the corrected complexity and add the corresponding scaling experiments.
>
> ---
>
> **[Q1] Meaning of “context” in Context-aware Role Generation**
>
> We thank the reviewer for the question and will add a clearer definition of “context” in the paper to avoid confusion. In our work, **“context”** refers to the **target variable-centric relational distribution observed within the current input window**, denoted as $e^{rel}_i$. Accordingly, “context-aware” means that, from the codebook representing all relation types, the model dynamically selects roles conditioned on the window- and variable-specific relational situation.
>
> ---
>
> **[Q2] How TS-TPR realizes “dynamic relation modeling”**
>
> We appreciate this concern and will clarify it in the manuscript. We agree that within a single input window, relation embeddings \\(e^{rel}\\) are fixed. Our claim of **“dynamic relation modeling”** refers to **how relations evolve across sliding windows and how they are represented/updated.** Prior methods also produce input-dependent attention/adjacency, but such dynamics are typically **continuous weight shifts without explicitly separating relation *types***, and relational patterns remain **entangled with temporal features.**
> In contrast, TS-TPR maps each window’s \\(e^{rel}_{i,j}\\) onto a reusable codebook of role bases, so the active relation types (roles) can explicitly switch across windows, while role-conditioned fillers are updated via binding/unbinding under the same role basis. This yields a **compositional form of dynamics**—relation ***types* are reused and *attributes* adapt per window**—supporting more stable and interpretable generalization under distribution shifts.

---

> > ### Comment · Reviewer_2Vwu · 2025-11-25
> >
> > Thank you for the additional experiments and clarifications. Some of my concerns regarding the ablation study and the overall pipeline have been partially addressed. **However, after carefully considering the explanations, I will maintain my original score.**
> >
> > The primary reason is that the connection to TPR remains conceptually weak. With substantial modifications applied to standard TPR operations in latent space, it is difficult to identify its core value. From my perspective, the proposed approach primarily relies on a set of complex attention mechanisms to combine inter-variable relational embeddings and temporal embeddings. In this regard, many alternative designs (e.g., treating one as query and the other as key/value) seem equally viable, and the necessity of the current binding/unbinding operations is not convincingly demonstrated.
> >
> > Additionally, I share the concerns raised by reviewers t4MD and sGA5 that claims—particularly regarding zero-shot capabilities, interpretability, and scalability—appear overstated given the presented evidence.
> >
> > I encourage the authors to evaluate the role and necessity of each architectural component more critically. For instance, whether the VQ mechanism is a principled requirement for enabling OOD robustness, or just used to reduce computational overhead.

---

### Official Review · Reviewer_t4MD · 2025-10-31

**Soundness:** 2
**Presentation:** 3
**Contribution:** 2
**Rating:** 2
**Confidence:** 5

**Summary:**

This paper introduces TS-TPR, a multivariate time-series forecasting approach that leverages Tensor Product Representations. The core idea is to model inter-variable relations as roles and temporal attributes as fillers. The authors design a two-stage filler computation process:
1. Relation-aware fillers: Uses attention to query source variables with relation embeddings.
2. Role-aware aggregation: Employs attention to query relation instances using target-specific role embeddings, generating role-aligned fillers for each target variable.

The forecasting mechanism relies on binding/unbinding operations implemented through linear attention, featuring a context-aware unbinding operator that considers the target's temporal context. The Role Selector utilizes a VQ codebook of relation prototypes to identify the top-M roles for each target variable. The authors demonstrate through ablation studies that both the Role Selector and the two-stage filler extraction are essential components.

**Strengths:**

1. Achieves state-of-the-art or near-state-of-the-art results on multiple benchmarks, with consistent gains across long-term, short-term, and zero-shot settings.
2. Decoupling relations (roles) from temporal attributes (fillers) and recombining them via a TPR-style binding/unbinding mechanism provides a principled way to model time-varying dependencies and explain which roles matter.
3. The hierarchical pipeline makes alignment between concrete relations and abstract roles explicit; ablation studies show consistent gains for the full combination.
4. Evaluation spans ETT, ECL, Weather, Traffic, and EPF datasets, including model complexity, training-time, and memory comparisons, strengthening the practical engineering case.

**Weaknesses:**

1. Equation 5 computes \(R_i\) based on overall distances \(D_{i,k}\) without time resolution. It is unclear whether roles adapt per timestep, per window, or remain static per series. Missing time-resolved role trajectories and stability metrics.
2. Claims of natural scaling to many variables are only demonstrated up to \(C=862\).
3. Most zero-shot transfers remain within the ETT-family; heterogeneous transfers would provide a stronger test of compositional generalization.
4. All benchmark datasets use data collected between 2011-2020 (too old).

**Questions:**

1. Are roles selected per window or per timestep, and can time-resolved role assignment trajectories be shown?
2. Can you provide training/inference curves versus \(C\) and evaluate sparse relation approximations?
3. How does hard VQ-based role selection compare to soft attention over the codebook or temperature-annealed VQ?
4. Can heterogeneous cross-domain transfers be added with leakage-safe normalization?

---

> ### Author Response · Authors · 2025-11-24
> **Rebuttal by Authors**
>
> Dear Reviewer t4MD,
>
> Thank you for your thoughtful reading of our manuscript and the constructive feedback you provided. We have carefully addressed each of your comments below and incorporated corresponding revisions into the updated manuscript.
>
> ---
>
> **[W1, Q1] Clarification on Context-aware Role Generation**
>
> Thank you for the comment. In Eq. (5), roles \\(R_i\\) are **selected per sliding window (not per timestep)** and **independently for each target variable \\(x_i\\)**. Specifically, we treat each input window as a single context, align all pairwise relation embeddings $e^{rel}_{i,j}$ observed in that window to the global codebook, and then, based on the target-centric context $e^{rel}_i$, dynamically select the top-\\(M\\) most consistent relation types as the role set \\(R_i\\) for each target variable. To improve conceptual clarity and readability, we directly incorporated the reviewer’s suggestion and revised **Section 3.3** with more explicit explanations.  In addition, **Figure 3** visualizes this window-wise dynamics: as the window slides over time, the same variable is mapped to different relation codes, revealing its role-assignment trajectory through code transitions.
>
> ---
>
> **[W2, Q2, W4] Dataset limitations.**
>
> Our main experiments are conducted on **widely adopted public benchmarks in time-series forecasting**, enabling fair and reproducible comparisons with recent baselines under the same protocol. Still, to address the reviewer’s recency concern, we are extending TS-TPR toward a foundation-model setting and running additional zero-shot evaluations on the post-2025 **Gift-eval** [1]. Since this extension requires sequential codebook training across multiple domains, it takes some time, but we will report the results in **Appendix B.5** once finalized. We strengthen TS-TPR by training the codebook across multiple domains, rather than only increasing \\(C\\) within one dataset, to accumulate recurring relation patterns and improve structural generalization. Accordingly, testing on the largest public benchmark (\\(C=862\\)) already offers a challenging large-scale setting that captures rich relations and validates scalability.
>
> In addition, while TS-TPR’s relation construction is theoretically \\(O(C^2)\\), real-world multivariate series exhibit strong relation sparsity. We therefore introduce **salient-relation filtering** and show that using only a subset of relations preserves performance while substantially reducing computation and memory, validating **sparse relation approximation** (please refer to our response to Reviewer sGA5 of W1). We will include the sparsification trade-off and \\(C\\)-scaling results in **Appendix B.2**.
>
> ---
>
> **[W3, Q4] Zero-shot evaluation limited to ETT family**
>
> We appreciate the reviewer’s common concern that our zero-shot generalization results are confined to the ETT family. Our consolidated explanation of this setting is provided in our response to Reviewer sGA5 ([W2, Q2]), and we kindly ask the reviewer to refer to that section.
>
>
> ---
>
> **[Q3] Comparison of alternative VQ methods**
>
> Thank you for the helpful suggestion. We added an ablation comparing four quantization strategies for mapping relation vectors into the codebook role space, and this will be reported in **Appendix B.1.**
> As shown in the table below, the Gumbel-Softmax approach proves to be the most stable overall, successfully combining the exploratory flexibility of soft methods with the discrete interpretability of hard methods. Accordingly, we adopt **Gumbel-Softmax VQ** as the final choice in our model, as it avoids premature discretization while producing well-separated relational role representations.
> - **Hard VQ + STE:** Discrete 1-of-K role assignment for clear and interpretable role categorization.
> - **Soft VQ:** Continuous weighted mixture over codebook entries for stable and smooth optimization.
> - **Mix (hard forward, soft backward):** Hard role usage with soft gradients for improved convergence stability.
> - **Gumbel-Softmax (temp annealed):** Temperature-controlled transition from soft exploration to hard discretization.
>
> | Mode | ETTh1 (MSE/MAE) | ETTh2 (MSE/MAE) | ETTm2 (MSE/MAE) | ETTm1 (MSE/MAE) |
> |---|---|---|---|---|
> | **hard** | 0.384 / 0.404 | 0.295 / 0.350 | 0.173 / 0.255 | 0.323 / 0.359 |
> | **mix** | 0.381 / 0.403 | 0.292 / 0.344 | 0.173 / 0.254 | 0.331 / 0.362 |
> | **soft** | 0.379 / 0.400 | 0.292 / 0.344 | 0.171 / 0.254 | 0.318 / 0.355 |
> | **gumbel** | **0.376 / 0.399** | **0.291 / 0.347** | **0.171 / 0.254** | **0.316 / 0.354** |
>
> ---
>
> _Reference_
>
> [1] Gift-eval: A benchmark for general time series forecasting model evaluation, Aksu et al., 2024. URL https://arxiv. org/abs/2410.10393 1145.3637528.3671926.

---

### Official Review · Reviewer_UUF1 · 2025-11-01

**Soundness:** 3
**Presentation:** 2
**Contribution:** 3
**Rating:** 6
**Confidence:** 4

**Summary:**

The paper addresses nonstationary inter-variable dependencies in multivariate forecasting by advocating a compositional approach that explicitly factorizes relational and temporal structure. It introduces TS-TPR, which uses tensor product representations: a context-aware role generator selects salient inter-series relationships at each time step, while a hierarchical filler extractor summarizes the corresponding temporal patterns; their tensor products yield explicit, structured, and scalable representations that adapt as dependencies shift. The authors claim state-of-the-art results across diverse benchmarks and highlight interpretability—via time-varying role/filler components—as a key advantage over methods that entangle relation and dynamics.

**Strengths:**

1. The target problem - multivariate time series forecasting is important, and it is interesting to see the proposed solution coming from the view of tensor product representation.
2. It's good to see that some empirical results on efficiency are provided. At the first glance, I have some doubts on how this method would affect the efficiency for both training and inference. Based on their results, I am convinced that the overhead is not significant.
3. The experiments include the zero-shot setting. I appreciate the results on the zero-shot setting as I believe that it can demonstrate the generalizability of the proposed method.

**Weaknesses:**

1. I think it might be better to include some experiments on synthetic datasets to directly support the claim "the framework not only adapts to new relational contexts but also provides interpretability by revealing which relationships guide each prediction". It would greatly improve the credibility.
2. The authors do not provide some experiments on how to balance the prediction accuracy and cookbook regularization. In Sec. 3.5, some values for $\alpha$ and $\beta$ are provided. However, I think it would be better to have some empirical results on the effect of different values.
3. The datasets used in the experiments, especially for zero-shot setting. There are several more comprehensive benchmarks proposed since 2025, e.g., fev-benchmark [1] and Gift-Eval [2]. I would suggest to have some results on those benchmarks and investigate how it compare with other baselines.

References:

[1] fev-bench: A Realistic Benchmark for Time Series Forecasting

[2] GIFT-Eval: A Benchmark For General Time Series Forecasting Model Evaluation

**Questions:**

1. Some typos in the manuscript. Please revise it. For example, ". urthermore," in Page 5.
2. I wonder if the authors can provide some case studies on Weather dataset to show that how the proposed method can resolve the distribution shift issue. As in Figure 1, the authors show that the relationship between two vars can vary across time. I believe that might be better to show some results directly on this.

---

> ### Author Response · Authors · 2025-11-25
> **Rebuttal by Authors**
>
> Dear Reviewer UUF1,
>
> We thank the reviewer for the thoughtful and constructive feedback. The points raised are very helpful for strengthening both the empirical support and the clarity of our claims. We plan to revise the paper as follows.
>
> ---
>
> **[W1,Q2] Additional case study**
>
> We agree that additional evidence would make our claims about relational adaptation and interpretability more convincing. In the revised version, we will add (i) a regime-shift synthetic multivariate experiment where ground-truth inter-variable relations change across regimes, showing that the learned codebook roles track these changes and indicate which roles are used at each prediction step, and (ii) a Weather case study that links the relation changes in Figure 1 to the evolution of codebook-role usage (in the style of Figure 3). Together, these studies will demonstrate how the framework adapts to new relational contexts under distribution shift and which relationships guide each forecast; they will be reported in **Appendix B.2**.
>
> ---
>
> **[W2] Ablation on codebook regularization balance \\((\alpha, \beta)\\)**
>
> We appreciate the reviewer’s concern about how to balance prediction accuracy and codebook regularization. In the revised version, we will clarify that we use a cosine schedule so that \\(\beta_t\\) gradually increases over training while \\(\alpha_t\\) remains relatively stable, allowing the model to first focus on forecasting accuracy and later emphasize codebook structure. In addition, we will include an ablation where \\(\alpha\\) is fixed and \\(\beta \in \\{0.0, 0.5, 1.0\\}\\). Across ETTh1/2, ETTm1/2, Traffic, ECL, and Weather, the resulting MSE/MAE remain nearly unchanged while code usage becomes more structured as \\(\beta\\) increases. This shows that the regularization operates in a stable regime where it does not harm accuracy but improves the relational organization and interpretability of the codebook; the detailed table and discussion will be added to the **Appendix B.2** and summarized in **Section 3.5**.
>
> | TS-TPR (T=96, F=96) | ETTh1        | ETTh2        | ETTm1        | ETTm2        | Traffic      | ECL          | Weather      |
> |---------------------|-------------|--------------|--------------|--------------|-------------|--------------|--------------|
> | Metric              | MSE / MAE   | MSE / MAE    | MSE / MAE    | MSE / MAE    | MSE / MAE   | MSE / MAE    | MSE / MAE    |
> | \(\beta = 0.0\)     | 0.378 / 0.400 | 0.286 / 0.338 | 0.319 / 0.357 | 0.172 / 0.255 | 0.434 / 0.274 | 0.145 / 0.247 | 0.157 / 0.205 |
> | \(\beta = 0.5\)     | 0.377 / 0.398 | 0.286 / 0.338 | 0.316 / 0.354 | 0.171 / 0.254 | 0.432 / 0.272 | 0.145 / 0.249 | 0.156 / 0.204 |
> | \(\beta = 1.0\)     | 0.377 / 0.398 | 0.286 / 0.338 | 0.316 / 0.354 | 0.171 / 0.255 | 0.437 / 0.272 | 0.145 / 0.249 | 0.157 / 0.205 |
>
> ---
>
> **[W3] Additional evaluation on recent benchmarks**
>
> We agree that **fev-bench** and **GIFT-Eval** are important benchmarks for evaluating the zero-shot generalization ability of foundation-level time-series models. Our main experiments focus on widely used **multivariate** benchmarks, which allow fair and reproducible comparison with recent baselines and directly test inter-variable relational modeling. Following the reviewer’s suggestion, we additionally test TS-TPR on the GIFT-Eval benchmark, but note two structural limitations: (i) a fully fair foundation-style zero-shot comparison would require careful multi-domain pretraining/adaptation of the codebook across many datasets, which involves substantial additional tuning and training time; we view this as an important direction for future work and will report preliminary multi-domain results as they become available, and (ii) many tasks in fev-bench/GIFT-Eval are **univariate or covariate-light**, where the relational advantages of TS-TPR cannot be fully exploited.
>
> This is reflected in the preliminary result on the univariate *saugeenday\_D* series:
>
> | Model         | saugeenday\_D (MSE / MAE) |
> |--------------|---------------------------|
> | PatchTST     | 1.113 / 0.535             |
> | iTransformer | 1.140 / 0.534             |
> | TimeXer      | 0.901 / 0.482             |
> | TS-TPR       | 1.116 / 0.532             |
>
> As shown in the table, in the univariate setting temporal-only baselines perform better, indicating that there is essentially no cross-variable structure for our relational module to leverage in this case. Nevertheless, we believe TS-TPR targets the **practically important multivariate scenarios**, where explicitly modeling inter-variable relations plays a key role in real-world applications. In future work, we plan to extend TS-TPR to a **multi-domain pretraining** setting (including fev-bench and GIFT-Eval) to learn **domain-shared relation types (codebook roles)** that can be **zero-shot transferred** to new domains—targeting zero-shot generalization not only at the level of marginal patterns, but at the level of **relational structure** itself.

---

### Meta-Review · Area_Chair_JvhU · 2026-01-06

**Summary:**

This paper proposes TS-TPR, a compositional forecasting model using tensor product representations, explicitly capturing dynamic inter-variable relationships in multivariate time series with improved accuracy and interpretability.

3 out of 4 reviewers are still negative towards this manuscript, even after rebuttal. I have carefully read the revised paper, the reviews, and the rebuttals, but still the following concerns make me difficult to recommend acceptance in the current form.

  (1) The model performance is somehow very limited. Especially on the Traffic dataset as shown in Table 8, it is much worse than iTransformer. The proposed method should have shown some advantages on datasets with larger variable like Traffic, but the model failed. Performance improvement on other datasets is also limited.

  (2) As Reviewer sGA5 stated, the model interpretability is overstated, not fully exploited, which makes the proposed methodology less convincing. Also, the authors fail to demonstrate how the proposed model achieve dynamic relation modeling as claimed.

  (3) As Reviewer 2Vwu stated, the motivation for introducing the Tensor Product Representation (TPR) is not so clear. How the proposed TPR improved the performance correspondingly.

Given these unresolved concerns regarding performance, interpretability, and methodological motivation, I recommend rejection.

**Reviewer Concerns:**

Most concerns from reviewers are still outstanding after the rebuttal.

**Reviewer Scores:**

I think all 4 reviewers would have kept their original score as 6, 2, 4, 4 after the rebuttal.

---

### Decision · Program_Chairs · 2026-01-26

Reject